# Contrasting effects of *Ksr2*, an obesity gene, on trabecular bone volume and bone marrow adiposity

**Gustavo A Gomez[1], Charles H Rundle[1,2], Weirong Xing[1,2], Chandrasekhar Kesavan[1,2], Sheila Pourteymoor[1], Robert E Lewis[3], David R Powell[4], Subburaman Mohan[1,2]***

[1]VA Loma Linda Healthcare System, Loma Linda, United States; [2]Loma Linda University Medical Center, Loma Linda, United States; [3]University of Nebraska Medical Center, Omaha, United States; [4]Lexicon Pharmaceuticals, The Woodlands, United States

**Abstract** Pathological obesity and its complications are associated with an increased propensity for bone fractures. Humans with certain genetic polymorphisms at the kinase suppressor of ras2 (KSR2) locus develop severe early-onset obesity and type 2 diabetes. Both conditions are pheno-copied in mice with *Ksr2* deleted, but whether this affects bone health remains unknown. Here we studied the bones of global *Ksr2* null mice and found that *Ksr2* negatively regulates femoral, but not vertebral, bone mass in two genetic backgrounds, while the paralogous gene, *Ksr1*, was dispensable for bone homeostasis. Mechanistically, KSR2 regulates bone formation by influencing adipocyte differentiation at the expense of osteoblasts in the bone marrow. Compared with *Ksr2*'s known role as a regulator of feeding by its function in the hypothalamus, pair-feeding and osteoblast-specific conditional deletion of *Ksr2* reveals that *Ksr2* can regulate bone formation autonomously. Despite the gains in appendicular bone mass observed in the absence of *Ksr2*, bone strength, as well as fracture healing response, remains compromised in these mice. This study highlights the interrela-tionship between adiposity and bone health and provides mechanistic insights into how *Ksr2*, an adiposity and diabetic gene, regulates bone metabolism.

## Editor's evaluation

This study represents an important advance in connecting bone biology and metabolic functions. It implicates *Ksr2* as a key regulator of the switch between adipocytes and osteoblasts that arise from a common precursor. Besides being an actionable target for obesity and osteoporosis, the study reaffirms and provides mechanistic data relating to the human genetic findings on KSR2 variants in metabolic regulation.

## Introduction

Obesity is a major public health problem in the United States, afflicting nearly 40% of adults, and has become a prevalent and destructive health disorder linked to some of the major metabolic diseases, including cardiovascular diseases, type 2 diabetes (T2D), and cancer (*Devlin and Rosen, 2015*; *Pagnotti et al., 2019*; *Shanbhogue et al., 2016*; *Walsh and Vilaca, 2017*). Although obesity may be considered beneficial to bone health, since increased body weight is associated with higher bone mineral density (BMD), the relationship between excess body fat and bone health is complex, given that obesity has been identified as a risk factor for certain fractures (*Greco et al., 2015*; *Ma et al.,*

*For correspondence:
Subburaman.Mohan@va.gov

**eLife digest** Our bones are living tissues which constantly reshape and renew themselves. This ability relies on stem cells present in the marrow cavity, which can mature into the various types of cells needed to produce new bone material, marrow fat, or other components.

Obesity and associated conditions such as type 2 diabetes are often linked to harmful changes in the skeleton. In particular, these metabolic conditions are associated with weight-bearing bones becoming more prone to facture and healing poorly. Mice genetically modified to model obesity and diabetes could help researchers to study exactly how these conditions – and the genetic changes that underlie them – impact bone health.

Gomez et al. aimed to address this question by focusing on *KSR2*, a gene involved in energy consumption and feeding behavior. Children who carry certain *KSR2* mutations are prone to obesity and type 2 diabetes; mice lacking the gene also develop these conditions due to uncontrolled eating.

Closely examining mutant mice in which *Ksr2* had been deactivated in every cell revealed that the weight-bearing bones of these animals were also more likely to break, and the fractures then healed more slowly. This was the case even though these bones had higher mass and less marrow fat compared to healthy mice. Non-weight bearing bones (such as the spine) did not exhibit these changes.

Further experiments revealed that, when expressed normally in the skeleton, *Ksr2* skews the stem cell maturation process towards marrow fat cells instead of bone-creating cells. This suggests a new role for *Ksr2*, which therefore seems to independently regulate both feeding behavior and bone health. In addition, the work by Gomez et al. demonstrate that *Ksr2* mutant mice could be a useful model to better understand how obesity and diabetes affect human bones, and to potentially develop new therapies.

*2018*; *Veldhuis-Vlug and Rosen, 2018*). The increasing prevalence of obesity and T2D dictates the need for appropriate in vivo animal models to study the mechanisms of action of obesity and T2D on bone metabolism.

The effect of obesity and T2D on bone is an active area of investigation. Studies with several animal models and approaches have contributed to our current understanding of this relationship. Mouse models, in particular, have provided invaluable information through controlled genetic, biochemical, cellular, and molecular approaches to understand the pathological relationship between excess body fat and bone fragility. Most diet-induced obesity studies have reported reduced BMD and trabecular bone mass (*Bonnet et al., 2014*; *Doucette et al., 2015*; *Inzana et al., 2013*; *Scheller et al., 2016*). By contrast, monogenetic models of obesity have provided a broader range of bone phenotypes, including no change, loss, or gain in bone mass or BMD (*Ahn et al., 2006*; *Baldock et al., 2007*; *Braun et al., 2012*; *Steppan et al., 2000*; *Wang et al., 2007a*). There are several explanations for the diversity in skeletal phenotypes in these models, including differences in expression of targeted genes in other tissues besides bone, as well as varied effects of endocrine factors produced in other affected tissues such as the brain, fat, and skeletal muscle. Nevertheless, monogenetic studies have informed the molecular underpinnings of feeding regulation at the hypothalamus, which has fortuitously led to the development of pharmaceuticals to treat a particular population of individuals genetically predisposed to diabetes (*Yeo et al., 2021*). Although the RANKL monoclonal antibody, denosumab, has been used to treat bone disorders in osteoporotic T2D patients with reduced BMD (*Abe et al., 2019*), whether these agents can also benefit the population with gains in BMD, which are paradoxically also fragile (*Burghardt et al., 2010*; *Ma et al., 2012*), remains to be investigated. Also, it is worthwhile to further identify/study animal models with genetic mutations that phenocopy the human condition to study these interventions. The advent of the genomic era has expanded the list of individual genes associated with obesity and T2D (*Loos and Yeo, 2022*), yet their effects on bone remain vastly understudied.

Recently, the scaffold proteins kinase suppressor of ras (KSR1 and KSR2) were identified as mediators of energy consumption, utilization, and adipogenic regulation (*Brommage et al., 2008*; *Costanzo-Garvey et al., 2009*; *Kortum et al., 2005*; *Pearce et al., 2013*; *Revelli et al., 2011*). Although these two genes function as paralogs, we previously found that only *Ksr2* knockout (KO) mice become

obese and diabetic (**Brommage et al., 2008**; **Costanzo-Garvey et al., 2009**; **Kortum et al., 2005**; **Pearce et al., 2013**; **Revelli et al., 2011**), suggesting that these paralogs have non-redundant roles, although *Ksr1* does have a role in adipogenesis (**Kortum et al., 2005**). Several mutations at the KSR2 loci in humans have been associated with severe early-onset obesity (**Pearce et al., 2013**), and studies in *Ksr2* KO mice have revealed a centrally regulated mechanism by *Ksr2* expression and function in the hypothalamus that results in hyperphagia, changes in metabolic rate, and consequently, obesity and T2D (**Costanzo-Garvey et al., 2009**; **Guo et al., 2017**; **Henry et al., 2014**; **Pearce et al., 2013**; **Revelli et al., 2011**). Although there are hundreds of mouse genes reported to lead to obesity when disrupted, *Ksr2* gene deletion is associated with a profound obese phenotype and lethality at a young age (**Brommage et al., 2008**; **Revelli et al., 2011**). In this study, we set out to investigate whether the deletion of *Ksr2*, an obesity and T2D gene, bears any effect on bone health, and if so, to evaluate the mechanisms by which KSR2 affects bone metabolism. Our studies show that loss of KSR2 function increases long bone trabecular bone mass while reducing marrow adiposity and that KSR2 acts as a molecular switch that controls the differentiation of bone resident mesenchymal stem cells into osteoblast or adipocyte differentiation via an mTOR-dependent mechanism.

## Results

### *Ksr2* negatively regulates femoral bone mass

To evaluate whether deletion of *Ksr2* affects skeletal morphology, femurs of *Ksr2* KO (exon 13 deleted) and wild-type (WT) control mice in the C57BL/6J-Tyr$^{c\text{-}Brd}$ × 129$^{SvEvBrd}$ hybrid background (**Figure 1A**) were subjected to micro-computed tomography (microCT) scanning. Distal femoral metaphyseal bones of *Ksr2* KO (*Ksr2$^{-/-}$*) female mice exhibited increased trabecular bone mass at both 11 and 15 weeks of age (**Figure 1B**). Quantification of trabecular parameters at the distal femur secondary spongiosa shows that by 11 weeks bone volume fraction (BV/TV), and trabecular thickness (Tb.Th) were significantly increased in female KO mice (**Figure 1C and G**), while structure model index (SMI), a measure of rod to plate-like trabecular morphology (**Hildebrand and Rüegsegger, 1997**), was significantly reduced (**Figure 1E**), overall implying that structural morphological changes elicited by the absence of *Ksr2* promote gains in bone mass. Although mean trabecular connectivity density (CONN.D) and trabecular number (Tb.N) were also increased in KO mice at 11 weeks (**Figure 1D and F**), these differences were not significant until 15 weeks, as was the reduction in trabecular spacing (**Figure 1H**).

New woven bone is actively formed and mineralized at the primary spongiosa while the woven bone is remodeled into mechanically stronger lamellar bone at the secondary spongiosa. To determine whether new bone formation at the primary spongiosa is altered in the *Ksr2* KO mice, we measured trabecular bone parameters at the primary spongiosa, limited to within 300 µm of the distal-most femoral metaphyseal bone from the growth plate (**Figure 1I–M**) and found significant increases in trabecular bone volume. *Ksr2* deleted males also exhibited significantly greater BV/TV, Conn. Den, Tb.N, and Tb.Th but reduced Tb.Sp and SMI compared to littermate control mice (**Figure 2A–G**). Thus, loss of the *Ksr2* gene promotes trabecular bone density in both genders of mice.

By further characterization of femoral bones, we found increased mid-shaft femoral cortical bones in *Ksr2* nulls. While total tissue volume, indicative of bone size, remained unchanged in the KO mice in both genders (**Figures 1N–P and 2H–K**), a significant increase in bone volume fraction was observed, although gains in BMD were more prominent in females. Nonetheless, this evaluation reveals that obese *Ksr2* null mice present increased gains in trabecular and cortical mass of femoral bones.

### *Ksr1* is dispensable for the development of femoral trabecular bone mass

*Ksr1*, the only paralog of *Ksr2*, is highly expressed in skeletal muscle (**Costanzo-Garvey et al., 2009**), which interacts with and affects bone physiology (**Lara-Castillo and Johnson, 2020**). *Ksr1* was also expressed in osteoblasts (data not shown). To determine whether *Ksr1* might also contribute to limb bone mass accretion, we evaluated metaphyseal femoral bones of 16-week-old *Ksr1* KO mice and their wild-type littermate controls by microCT. In contrast with the striking differences found in trabecular bone parameters in *Ksr2* KO mice at this age, trabecular bone measurements were nearly identical

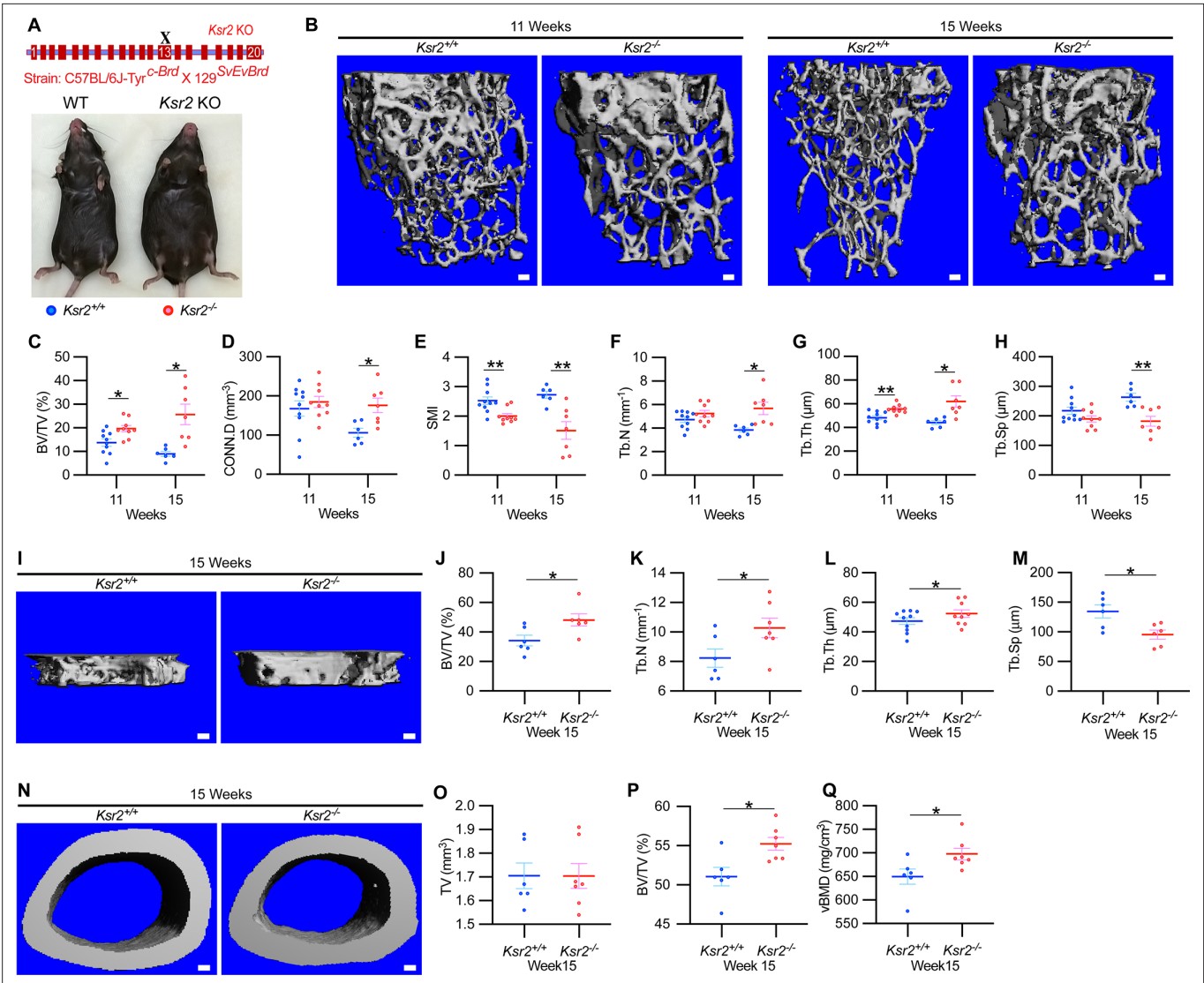

**Figure 1.** *Ksr2* regulates bone mass in females. (**A**) Schematic of *Ksr2* knocked out in the C57BL/6J-Tyr*c-Brd* × 129*SvEvBrd* hybrid strain with exon 13 deleted (X), and accompanying ventral view of genotyped mice at 4 months of age exhibiting differences in weight gain. (**B**) Representative 3D micro-computed tomography (microCT) reconstruction images of the secondary spongiosa at the distal femoral metaphysis in wild-type (*Ksr2+/+*, WT) or knockout (*Ksr2-/-*, KO) females at 11 and 15 weeks, revealing a prominent increase in trabecular bone in KOs. Scale bar: 100 µm. (**C–H**) MicroCT measurements from the trabecular bone as represented in panel (**B**) (n = 6–10/group), BV/TV, bone volume/tissue volume; CONN.D, connectivity density; SMI, structural model index; Tb.N, trabecular number; Tb.Th, trabecular thickness; Tb.Sp, trabecular spacing. (**I**) Representative 3D reconstruction of microCT images of primary spongiosa in WT or KO mice at 15 weeks of age revealing increased bone density in KO mice. Scale bar: 100 µm. (**J–M**) Quantification of microCT parameters measured in panel (**I**) (n = 6–10/group). (**N**) Representative 3D reconstruction of microCT images of cortical bone at the femoral mid-diaphysis (scale bar: 100 µM), where the TV total volume (**O**) is not affected, while BV/TV and volumetric bone mineral density (vBMD) (**P, Q**) are increased in KO mice. Statistics analyzed by unpaired two-tailed Student's *t*-test, and graphed lines represent the mean ± SEM, *p<0.05, **p<0.005.

The online version of this article includes the following source data for figure 1:

**Source data 1.** Micro-computed tomography (microCT) measurements of female trabecular bone.

**Source data 2.** Micro-computed tomography (microCT) measurements of female primary spongiosa.

**Source data 3.** Micro-computed tomography (microCT) measurements of female cortical bone.

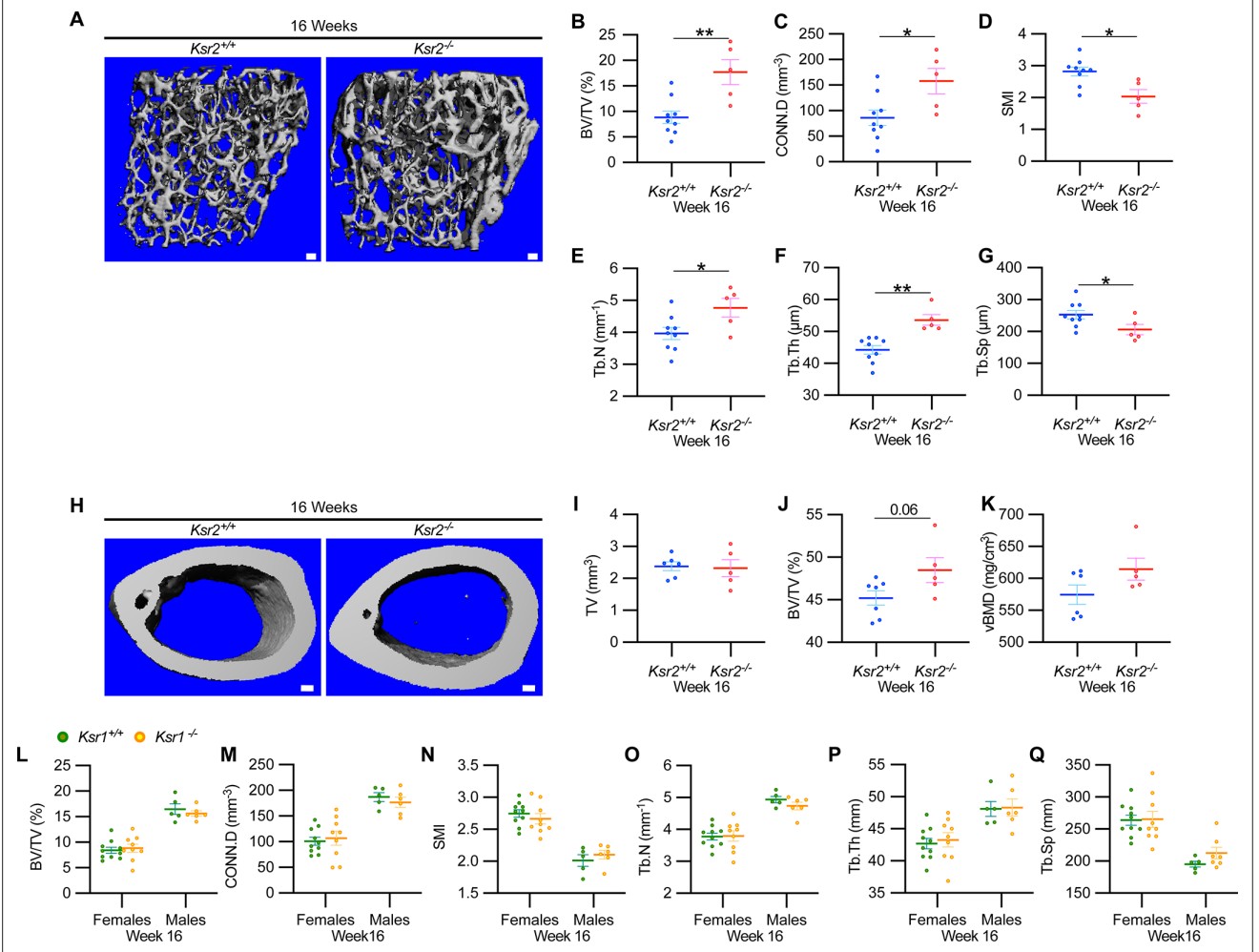

**Figure 2.** *Ksr2* negatively regulates femoral bone in males, while *Ksr1* deletion does not affect trabecular bone in either gender. (**A**) Representative 3D micro-computed tomography (microCT) reconstruction images of the distal femoral metaphysis in wild-type (WT) or knockout (KO) male mice at 16 weeks of age revealing increased trabecular bone in KOs. Scale bar: 100 μm. (**B–G**) MicroCT measurements from the trabecular bone as represented in panel (**A**) (n = 5–9 mice per group). BV/TV, bone volume/tissue volume; CONN.D, connectivity density; SMI, structural model index; Tb.N, trabecular number; Tb.Th, trabecular thickness; Tb.Sp, trabecular spacing. (**H**) Representative 3D microCT reconstruction images of cortical bones at the femoral mid-diaphysis revealing that *Ksr2* deletion does not affect TV total volume (**I**) of cortical bone in males, while BV/TV (**J**) and volumetric bone mineral density (vBMD) (**K**) are increased in Ksr2 KO mice (n = 5–6/group). Scale bar: 100 μm. (**L–Q**) Quantification of microCT data from the distal femoral metaphysis of WT and *Ksr1* knockout mice at 16 weeks of age, showing minimal changes in trabecular bone parameters between genotypes in either gender. Statistics analyzed by two-tailed Student's *t*-test, and graphed lines represent the mean ± SEM, *p<0.05, **p<0.005.

The online version of this article includes the following source data for figure 2:

**Source data 1.** Micro-computed tomography (microCT) measurements of male trabecular bone.

**Source data 2.** Micro-computed tomography (microCT) measurements of male cortical bone.

**Source data 3.** Micro-computed tomography (microCT) measurements of *Ksr1* knockout mice.

between *Ksr1* KO and WT mice, irrespective of gender (*Figure 2L and M*). These results suggest that the KSR1 protein is highly unlikely to synergize with KSR2 in regulating femoral bone growth.

## Validation of bone phenotype by *Ksr2* deletion in a different genetic background

Since genetic background can influence biological effects in mice (*Ackert-Bicknell et al., 2009*; *Bonnet et al., 2014*), we evaluated whether *Ksr2* deleted in the DBA/1LacJ strain, which also becomes obese (*Costanzo-Garvey et al., 2009*), might also exhibit alterations in skeletal phenotype. Deletion of exon 4 in this genetic background did not significantly affect the anus to nose body length, although a

substantial gain in body weight and percent body fat was observed by 8 weeks of age (*Figure 3A–D*). Concordant with *Ksr2* deletion in the C57BL/6J-Tyr$^{c\text{-}Brd}$ × 129$^{SvEvBrd}$ hybrid background, dual-energy X-ray absorptiometry (DXA) measurements of pooled genders revealed substantial increases in total body BMD (*Figure 3E*) and, in particular, in femoral BMD (*Figure 3G*), while femur length remained unchanged relative to WT siblings (*Figure 3F*). Additionally, no changes were observed in the vertebral trabecular bone (*Figure 3—figure supplement 1*).

Histological evaluation of longitudinal distal metaphyseal femur bone sections by alizarin red also revealed increased amounts of calcified bone in *Ksr2* KO mice as the ratio of bone area over total area was increased, while osteoid area over bone area was decreased (*Figure 3H–J*, *Figure 3—figure supplement 2*). Moreover, qualitative comparisons of bone markers, integrin bone sialoprotein (IBSP) and secreted phosphoprotein 1 (SPP1), by immunofluorescence suggest an increased areal expansion of both markers throughout the metaphysis in KO mice (*Figure 3Q and R*). Overall, this data provides further supporting evidence that *Ksr2* negatively regulates appendicular bone formation, with confirmation in a different genetic background.

## Gains in bone mass in *Ksr2* nulls are a product of increased osteoblast activity

Next, we began to address how the deletion of *Ksr2* results in increased bone mass. To determine whether increased bone formation is the cause of increased bone mass in *Ksr2* KO mice, we performed histomorphometric analysis by pulsed calcein injections in 8-week-old mice. This resulted in increased calcein labeling in KO mice, with quantitative gains detected in bone formation rate and mineral apposition rate (*Figure 3K–M*). By contrast, the percentage of bone-resorbing acid phosphatase 5, tartrate-resistant (ACP5+) osteoclasts scored per bone surface, did not change (*Figure 3N*). Moreover, serum levels of osteoblast bone deposition (procollagen type 1 N-terminal propeptide [P1NP]) were elevated in KO mice, while those of osteoclast activity (carboxy-terminal cross-linked telopeptide of type 1 collagen [CTX1]) were not changed (*Figure 3O and P*).

Bulk comparisons between osteoblast and osteoclast markers were then compared by real-time-quantitative PCR (RT-qPCR) from the metaphysis of *Ksr2* KO relative to WT littermates at 12 weeks of age. mRNA expression levels of osteoblast markers, *Alpl* and *Bglap2,* showed increased, though insignificant, expression in KO mice, while *Spp1* and *Sp7* were significantly increased. By contrast, markers of differentiated osteoclasts, *Acp5* and *Ctsk*, remained unchanged (*Figure 3S*). Combined, these results posit that *Ksr2* affects osteoblast function, and does not apparently affect osteoclasts.

## *Ksr2* gains bone at the expense of adipocyte differentiation

As *Ksr2* KO mice are obese, exhibiting increased visceral and subcutaneous adiposity (*Figure 4A*), we determined whether genetic disruption of *Ksr2* influences adipocyte gene expression in white and brown adipose tissues in 28-week-old mice. mRNA levels of key transcription factors, *Pparg* and *Cebpa,* were unchanged in both fat depots in *Ksr2* KO mice (*Figure 4B*) at this age. By contrast, the adipokine leptin (*Lep*) was increased, while complement factor D (*Cfd*) was decreased in both white and brown fat of *Ksr2* KO mice (*Figure 4C*). This suggests that KSR2 exerts opposite effects on leptin and complement factor D/adipsin expression in fat tissues, which is consistent with changes observed in other models of obesity in mice (*Cianflone et al., 2003*; *Kwon et al., 2012*).

Bone marrow stem/stromal cells (BMSCs) represent common precursors for adipocytes and osteoblasts, and marrow adipose tissue (MAT) volume is known to be inversely correlated with trabecular bone mass (*Ko et al., 2021*; *Pierce et al., 2019*; *Tencerova et al., 2018*; *Yue et al., 2016*). To determine whether MAT volume is affected in the *Ksr2* KO mice, we evaluated the levels of osmium tetroxide retaining MAT in femurs of 28-week-old mice by microCT (*Figure 4D*). MAT volume was significantly reduced in all three compartments (proximal and distal metaphysis, and diaphysis) in the tibia of *Ksr2* KO mice compared with controls (*Figure 4E*). Consistent with these data, there was a significant reduction in adipocytes in the distal femoral metaphysis of *Ksr2* KO mice (*Figure 4 F and G*).

To determine the potential regulatory molecules that contribute to changes in MAT in *Ksr2* KO mice, we compared mRNA expression of adipocyte markers at the trabecular compartment in distal femurs of *Ksr2* KO and control mice. Neither markers associated with white adipocytes (*Fabp4*, *Slc7a10*) or brown adipocytes (*Ucp1*, *Prdm16*) were different in *Ksr2* KO femurs, while of the adipogenic

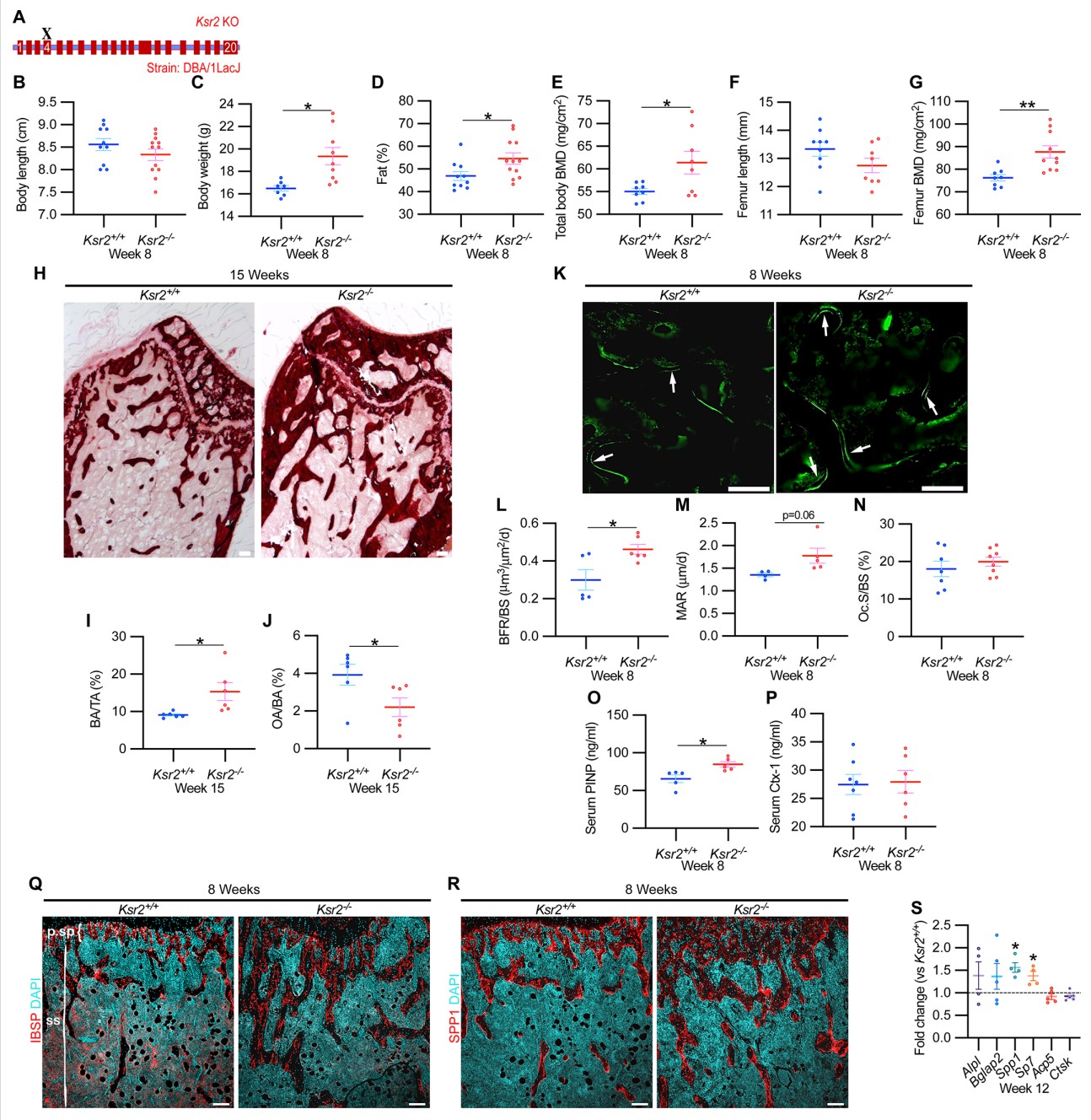

**Figure 3.** *Ksr2* deletion in a different genetic background, histomorphometry, and histology validates that *Ksr2* negatively regulates bone formation. (**A**) Schematic of *Ksr2* knocked out in the DBA/1LacJ strain with exon 4 deleted (X). (**B**) No differences were noted in body length at 8 weeks of age, while gains in body weight (**C**) and body fat percentage (**D**) are noted in knockouts (KOs). Bone mineral density (BMD) is increased in total body (**E**) and femurs (**G**) of KO mice, while femur length is not changed (**F**) (n = 7–12 mice/group; genders combined) (**D–G** reflect dual-energy X-ray absorptiometry measurements). (**H**) Representative alizarin red images at the distal femoral epiphysis show increased area of mineral staining in KO mice at 11 weeks of age. Scale bar: 100 μm. (**I, J**) Quantification of alizarin stain reveals an increase in bone area/total area (BA/TA) and a decrease in osteoid area/ bone area (OA/BA). (**K**) Representative histomorphometric images of fluorescent calcein label reveal increased staining in KO mice. Scale bar: 100 μm. (**L–N**) Quantification of histomorphometric parameters measured, showing increased bone formation rate/bone surface (BFR/BS) and mineral apposition rate (MAR), yet no changes in the number of osteoclasts per bone surface (Oc.S/BS) (n = 4–7 mice/group). (**O, P**) Serum levels of bone formation marker (PINP) and bone resorption marker (Ctx-1) in 8-week-old female *Ksr2* mutant and wild-type mice (n = 5–7 mice/group). (**Q, R**) Immunofluorescence staining at distal femoral metaphysis for (IBSP, synonym BSP2) or (SPP1, synonym OPN) (both red), counterstained with DAPI (cyan) reveals broader expression of both bone markers in KO mice; growth plate-osteoblast boundary positioned at the top. p.sp, primary spongiosa; ss, secondary spongiosa. Scale bar: 100 μm. (**S**) RT-qPCR reveals increased expression of osteoblast markers (*Alpl*, *Bglap2*, *Spp1*, and *Sp7*), while osteoclast markers

*Figure 3 continued on next page*

*Figure 3 continued*

(*Acp5, Ctsk*) remain unchanged in femurs of KO mice. Statistics analyzed by two-tailed Student's *t*-test, and graphed lines represent the mean ± SEM, *p<0.05, **p<0.005.

The online version of this article includes the following source data and figure supplement(s) for figure 3:

**Source data 1.** X-ray measurements of *Ksr2* knockouts in DBA/1LacJ.

**Source data 2.** Distal femur–alizarin red quantification.

**Source data 3.** Histomorphometric measurements of *Ksr2* knockout mice.

**Source data 4.** Serum ELISA measurements of *Ksr2* knockout mice.

**Source data 5.** RT-qPCR data of 12-week-old *Ksr2* knockout femurs versus wild-type.

**Figure supplement 1.** Vertebral trabecular bone is not affected by *Ksr2*.

**Figure supplement 1—source data 1.** Micro-computed tomography (microCT) measurements of lumbar vertebrae.

**Figure supplement 2.** Osteoid area is regulated by *Ksr2*.

regulatory transcription factors evaluated, *Pparg* was mildly but significantly reduced, and *Cebpa* was not changed. However, key adipokines (*Lep, Cfd*) were decreased (*Figure 4H*). Since Wnt signaling is critically involved in regulating adipocyte differentiation, we also measured mRNA levels of several Wnt-related genes, but only found a decrease in *Wnt8b*, and an increase in *Ccnd1*, a Wnt target gene, in the bones of *Ksr2* KO mice (*Figure 4I*). Thus, changes in adipokine gene expression in both body and MAT adipocytes are altered when *Ksr2* is deleted globally.

## Loss of *Ksr2* delays femoral fracture healing and results in more fragile bones

The pathological obese/T2D condition predisposes bones to compromised fracture healing and increased fracture risks. Since the absence of *Ksr2* results in increased appendicular bone deposition, we evaluated whether this increased rate of bone formation would prove beneficial in *Ksr2* KO mice. Healing response was compared between WT and KO mice at 16 weeks of age following stabilized closed femoral fractures (*Figure 5A*). X-ray analysis 3 weeks after fracture showed improved bony union of the callus in WT mice and increased callus size in KO mice (*Figure 5B*).

MicroCT measurements also showed increased total volume in the fracture callus of KO mice (*Figure 5C and D*). When the callus was segmented into low and high densities for analysis, an increase in low-density woven bone volume was observed in *Ksr2* KO fracture callus but no changes were observed in high-density cortical bone volume (*Figure 5E and F*). Consequently, the low-density callus BV/TV in the *Ksr2* KO fractures was not significantly different from WT, while the high-density BV/TV callus was reduced in *Ksr2* KO mice (*Figure 5E and F*). Therefore, KO mice exhibited a larger fracture callus with increased low-density woven bone and reduced high-density cortical bone, consistent with delayed fracture callus development.

Histological evaluation also revealed increased Safranin O-stained cartilage in *Ksr2* KO mice but no difference in TRAP+ osteoclasts compared to WT mice (*Figure 5G–J*), suggesting that the differences observed between WT and KO were not caused by differences in bone resorption, but rather by delayed endochondral ossification. Proteins associated with hypertrophic chondrocytes, COL10A1 and IBSP (*Gomez et al., 2022*), were also increased in *Ksr2* KO mice, while SP7 was reduced, indicating that the delay in bone formation occurred during the ossification of the hypertrophic cartilage (*Figure 5K–N*). Combined, these results show that although deletion of *Ksr2* leads to obesity with increased bone mass, fracture healing is compromised despite the increased bone accretion observed in unfractured bones.

Since increased trabecular bone mass associated with obesity/T2D in humans is paradoxically associated with an increased risk of fracture (*Greco et al., 2015*; *Ma et al., 2018*; *Moseley, 2012*; *Oei et al., 2013*), femoral bones of WT and KO mice were tested for resistance to fracture by a three-point bending test. A lighter load was required to break KO bones, which displayed reduced stiffness, yet no change in elasticity (*Figure 5O–Q*). Therefore, as with humans, the femoral bones of *Ksr2* KO mice likely have structural integrity deficits that are more prone to fracture.

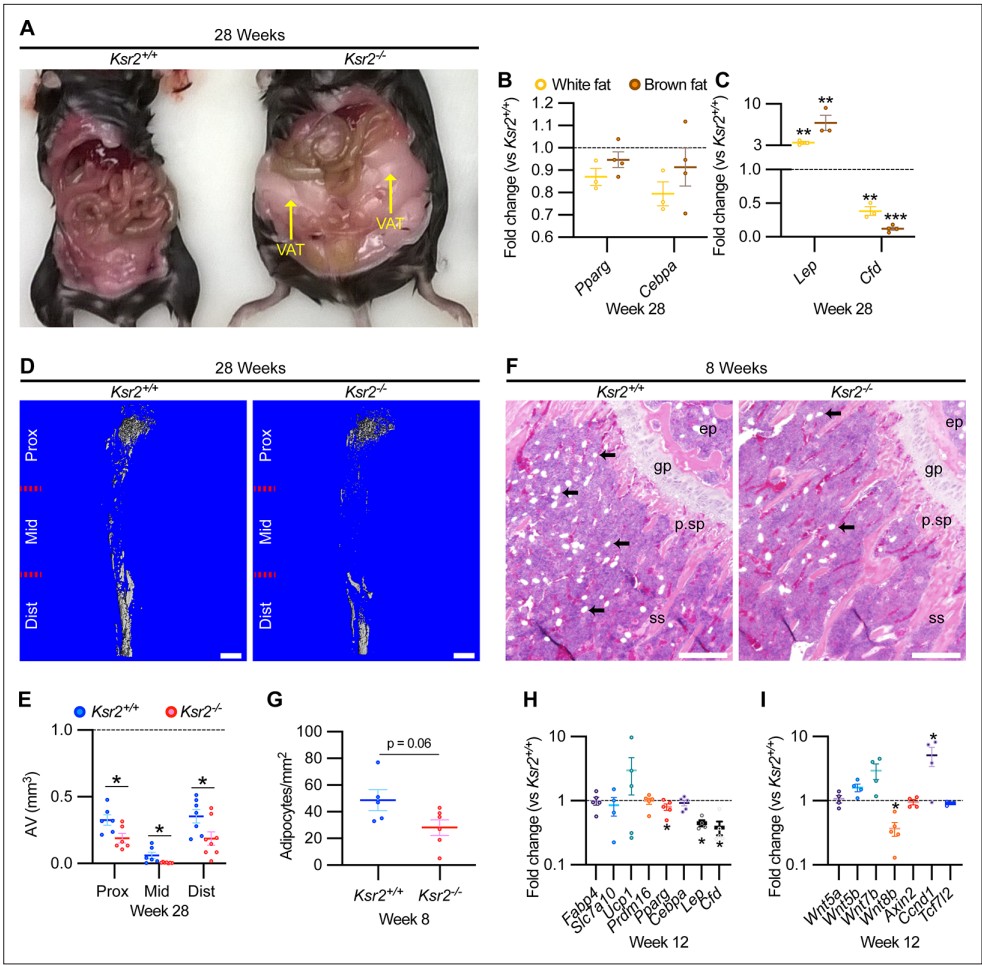

**Figure 4.** Obesity in *Ksr2* null mice paradoxically presents reduced bone marrow adiposity. (**A**) Representative image of mice at 28 weeks of age dissected to reveal differences in visceral adipose tissue (VAT) between wild-type (WT) and *Ksr2* knockouts (KOs). (**B, C**) RT-qPCR assessing changes in regulators of adipogenesis (**B**) or adipokine genes (**C**), in white or brown fat of *Ksr2* KO mice relative to WT at 28 weeks of age (n = 3–4/group). (**D**) Representative 3D micro-computed tomography (microCT) reconstruction images of osmium tetroxide-labeled femurs, revealing reductions in bone marrow adipose tissue in *Ksr2* KO mice at 28 weeks of age. Scale bar: 1 mm. (**E**) Quantification of adipocyte volume (AV) occupied by marrow adipose tissue in femurs of mice as depicted in panel (**D**), at proximal (prox), middle (mid), and distal (dist) thirds of the femur with position defined in reference to the spinal cord (n = 6–8/group). (**F**) Representative hematoxylin and eosin-stained longitudinal distal femur sections of 8-week-old mice in which adipocytes (arrows) were compared at the secondary spongiosa, revealing reductions in KO mice. Scale bar: 100 μm. (**G**) Quantification of sections as represented in panel (**F**) (n = 5–7/group). RT-qPCR comparisons in adipogenic (**H**) and Wnt-related (**I**) genes from the secondary spongiosa of femurs as shown in (**F**) (n = 3–5/group). Statistics analyzed by two-tailed Student's *t*-test, and lines plotted reflect the mean ± SEM, *p<0.05, **p<0.005, ***p<0.0005.

The online version of this article includes the following source data for figure 4:

**Source data 1.** RT-qPCR data of 28-week-old *Ksr2* knockout versus wild-type (adipose tissue).

**Source data 2.** Quantification of osmium tetroxide-labeled micro-computed tomography (microCT) of tibia.

**Source data 3.** Quantification of adipocytes from H&E-stained femurs metaphysis.

**Source data 4.** RT-qPCR data of 12-week-old *Ksr2* knockout versus wild-type femur (adipocyte markers).

**Source data 5.** RT-qPCR data of 12-week-old *Ksr2* knockout versus wild-type femur (Wnt markers).

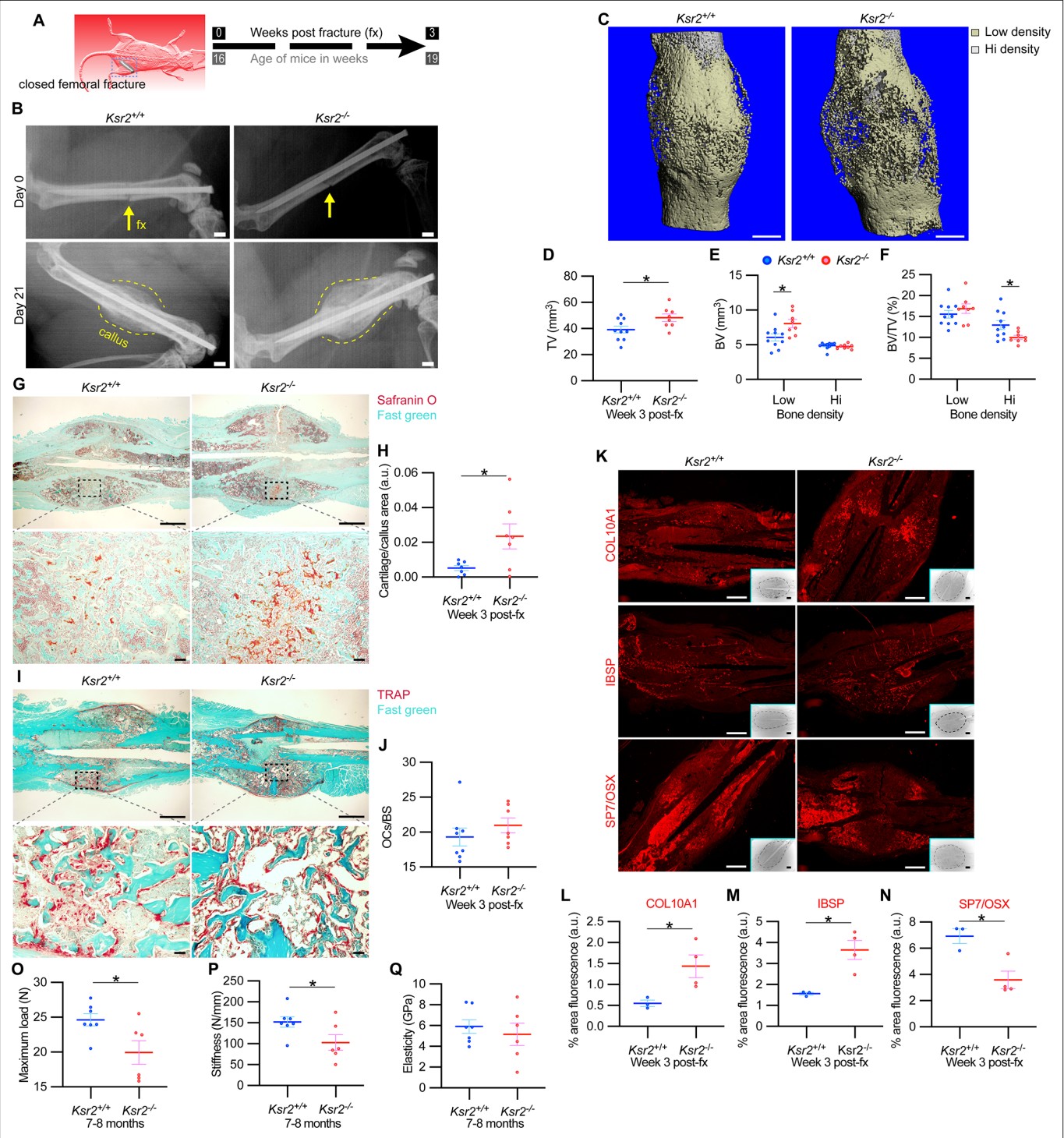

**Figure 5.** Delayed fracture healing but increased fragility in *Ksr2* knockout mice. (**A**) Schematic of strategy. (**B**) Representative X-ray images of bones that underwent closed mid-femoral fracture in wild-type (WT) and *Ksr2⁻/⁻* knockout (KO) mice on the day of surgery (day 0) and day 21 post-fracture (fx). Yellow arrows point to induced fracture, while calluses are outlined by dotted yellow lines. Scale bar: 1 mm. (**C**) Representative 3D micro-computed tomography (microCT) reconstruction images of fracture calluses at 3 weeks post-fx. Color-coded differences in bone density are indicated in the legend. (**D–F**) Quantification of microCT data for total volume (TV), bone volume (BV), and BV/TV (n = 8–10/group). (**G**) Representative Safranin O-stained chondrocytes in WT and *Ksr2⁻/⁻* bones at 3 weeks post-fx showing increased cartilage in KO mice and corresponding quantification (**H**) (n = 6–7/group). (**I**) Representative TRAP-stained osteoclasts in calluses at 3 weeks post-fx., showing no difference between genotypes, with corresponding quantification of osteoclasts/bone surface within callus (**H**) (n = 7–8/group). All histology sections were counterstained with fast green dye. Scale bars: 1 mm at low magnification (mag), top rows; or 100 µm at high magnification, bottom rows. (**K**) Representative immunofluorescence images for

*Figure 5 continued on next page*

*Figure 5 continued*

COL10A1, IBSP, SP7/OSX of fracture callus at 3 weeks post-fx. Dashed lines within the insets delineate the callus area quantified. Scale bar: 1 mm at low-magnification insets; 100 µm at high magnification. (**L–N**) Quantitation of fracture callus. (**O–Q**) Three-point bending test shows femurs of *Ksr2*^-/- KO mice tolerate less load to fracture with reduced stiffness, while elasticity remains unchanged. N, Newton; GPa, GigaPascal (n = 6–7/group mixed genders; two males per group). Statistics were analyzed by two-tailed Student's *t*-test, and graphed lines represent the mean ± SEM, * p<0.05.

The online version of this article includes the following source data for figure 5:

**Source data 1.** Micro-computed tomography (microCT) measurements of fracture callus after 3 weeks.

**Source data 2.** Quantification of Safranin O and ACP5/TRAP of fracture callus after 3 weeks.

**Source data 3.** Quantification of immunofluorescence images of fracture callus after 3 weeks.

**Source data 4.** Quantification of three-point bending tests.

## *Ksr2* is sufficient to inhibit osteoblast but not osteoclast differentiation

While *Ksr2* was known to be highly expressed in the brain, its expression in bone was unknown. By immunofluorescence using longitudinal sections of distal femoral bone sections from 3-week-old WT mice, positive KSR2 staining was observed at the epiphyseal secondary ossification center, and the metaphyseal region encompassing the primary and secondary spongiosa, coinciding with SPP1 (*Figure 6A*). To further explore whether *Ksr2* is expressed in osteoblasts and/or osteoclasts, we evaluated whether *Ksr2* mRNA is expressed during osteoblast or osteoclast differentiation from primary pre-osteoblasts or macrophages isolated from WT calvarial or femoral bones, respectively. The fidelity of differentiation in each condition was reflected by temporal upregulation of *Alpl* mRNA in osteoblasts or *Acp5* mRNA in osteoclasts, relative to vehicle-treated controls (*Figure 6B and C*). In osteoblasts, *Ksr2* exhibits a biphasic response, being inhibited on day 3 and transiently upregulated on day 14, before returning to basal levels on day 21, while *Ksr1* levels hovered around the baseline (*Figure 6B*). By comparison, *Ksr2* was only upregulated at the end of differentiation in osteoclasts (*Figure 6C*). Combined, this data shows that KSR2 is expressed in osteoblasts in vivo, and during osteoblast and osteoclast differentiation ex vivo.

Therefore, we assessed whether Ksr2 has any effect on the differentiation of either of these two lineages by a gain-of-function approach. BMSCs or macrophages isolated from femurs of 3-week-old WT mice were transduced with lentiviral vectors encoding either the KSR2 open-reading frame, or GFP as controls, and evaluated for differences in differentiation potential. BMSCs harboring GFP or overexpressed (OE) KSR2 (10–30-fold) underwent osteoblast differentiation in the presence of ascorbic acid (AA) or vehicle for 7 days. Alizarin red staining shows reduced differentiation in KSR2 OE cultures compared to GFP controls (*Figure 6D*). Ksr2 OE also reduced the expression of osteoblast differentiation markers *Alpl*, *Ibsp*, and *Spp1*, as well as *Runx2* and *Sp7*, while *Ccnd1* was not changed (*Figure 6E and F*), suggesting KSR2 exerts direct effects on osteoblast differentiation.

Macrophage differentiation toward the osteoclast lineage was achieved in both GFP and KSR2 OE cells as noted by the presence of TRAP-stained multinuclear osteoclasts in both populations on day 6 (*Figure 6G*), and their quantification resulted in minimal differences (*Figure 6H*). Also, we found no changes in *Acp5* or *Ctsk* between these cells, while *Ksr2* continued to be overexpressed (*Figure 6I*). Thus, this gain-of-function strategy suggests that KSR2 negatively regulates osteoblast differentiation from BMSCs but is likely dispensable for osteoclast differentiation.

## *Ksr2* regulates bone formation autonomously

To formally address whether central hypothalamic KSR2-mediated obesity might also regulate distal limb bone formation nonautonomously, we took a two-pronged approach. In the first approach, *Ksr2* KO mice were split into two groups, one was allowed to feed ad libitum (Ad lib), while the other group was pair-fed according to amounts eaten by WT mice for 12 consecutive weeks starting at 4 weeks after birth. As reported previously (Revelli et al.), *Ksr2* KO mice consumed twice as much food on average compared to WT mice. Also, serum leptin levels were increased several fold in the Ksr2 KO mice, which was rescued by pair-feeding according to amount of food eaten by WT control mice (*Revelli et al., 2011*). However, serum leptin levels were not measured in this study. Ad lib-fed KO mice again showed significant gains in BW, percent body fat, and femoral BMD relative to WTs, while pair-fed KO mice neither gained weight nor body fat but retained the increased femoral BMD

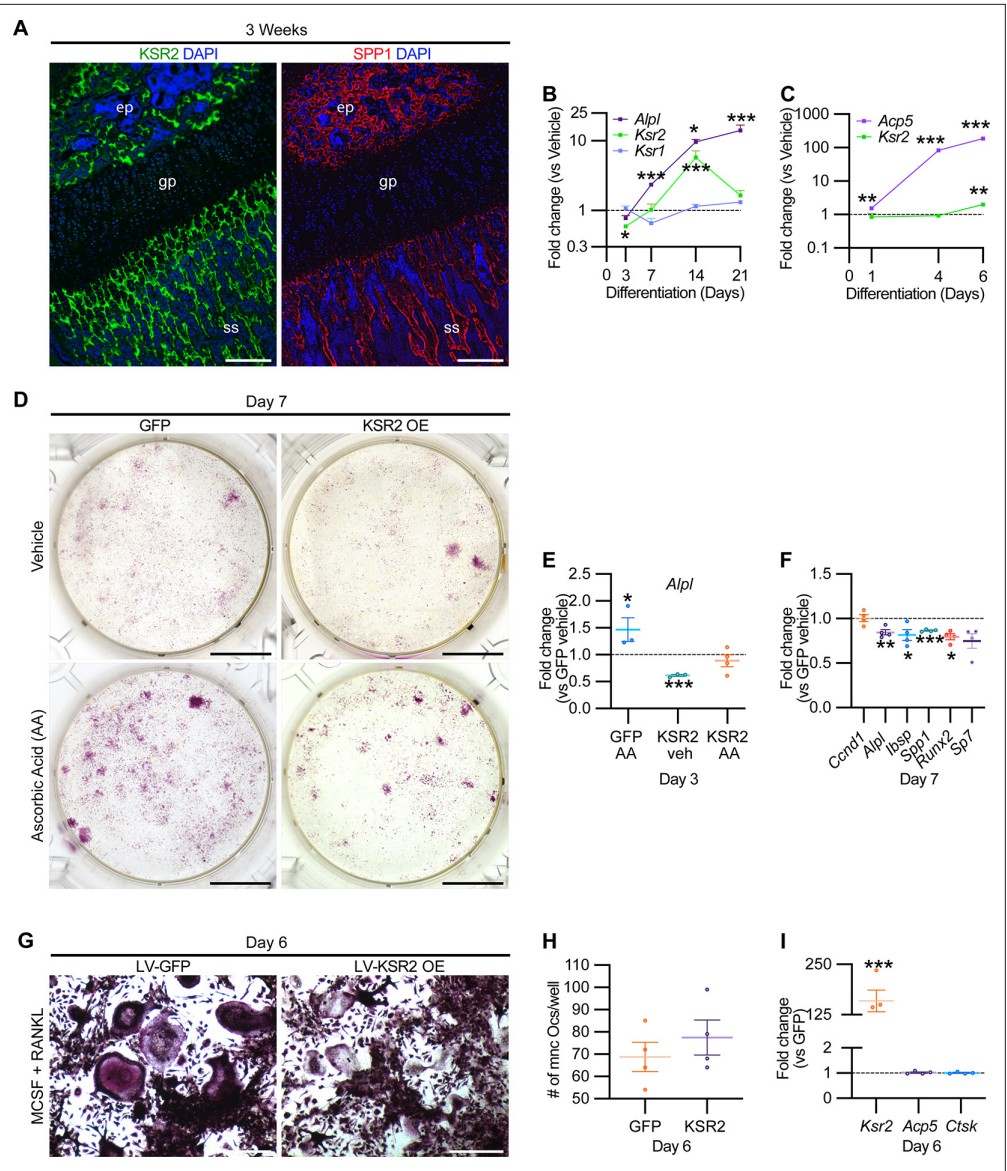

**Figure 6.** KSR2 is expressed in bone, and ex vivo gain-of-function studies demonstrate *Ksr2* represses osteoblast differentiation but is dispensable for osteoclast differentiation. (**A**) Representative immunofluorescence for KSR2 (green) and OPN (red) expression in longitudinal sections of distal femur epiphysis from 3-week-old wild-type (WT) mice, counterstained with DAPI (blue). ep, epiphyseal bone; gp, growth plate; ss, secondary spongiosa. Scale bar: 100 μm. (**B**) Ex vivo time-course RT-qPCR characterization of *Alpl*, *Ksr2*, and *Ksr1* mRNA expression on calvaria pre-osteoblasts isolated from WT mice following induction with osteoblast differentiation conditions relative to vehicle treatment (n = 3–4 independent experiments). (**C**) Ex vivo time-course RT-qPCR characterization of *Acp5* and *Ksr2* on primary macrophage cells isolated from femoral bones of WT mice following osteoclast differentiation relative to vehicle treatment (n = 3–4 independent experiments). (**D**) Representative images of alizarin red-stained primary bone marrow stromal cells with forced expression of either GFP or KSR2 after 7 days of treatment with either vehicle or ascorbic acid (AA). Scale bar: 10 mm. (**E**) RT-qPCR for *Alpl* on day 3 or (**F**) various osteoblast markers on day 7 (n = 4 independent experiments). (**G**) Representative images of multinuclear osteoclasts stained with ACP5/ TRAP following 6 days of osteoclast differentiation from macrophages isolated from femurs and transduced with either GFP or KSR2. Scale bar: 100 μm (**H**) Quantification of multinuclear osteoclasts (Ocs) counted/well as shown in panel (**G**). (**I**) RT-qPCR for *Ksr2*, or osteoclast markers *Acp5*, *Ctsk* in osteoclasts on day 6 of differentiation as represented in panel (**G**). Statistics analyzed by two-tailed Student's *t*-test, and graphed lines represent the mean ± SEM, *p<0.05, **p<0.005, ***p<0.0005.

The online version of this article includes the following source data for figure 6:

*Figure 6 continued on next page*

*Figure 6 continued*

**Source data 1.** RT-qPCR data of ex vivo osteoblast differentiation time course.

**Source data 2.** RT-qPCR data of ex vivo osteoclast differentiation time course.

**Source data 3.** RT-qPCR data for osteoblast differentiation from bone marrow stem/stromal cells (BMSCs).

**Source data 4.** RT-qPCR data for osteoblast differentiation from bone marrow stem/stromal cells (BMSCs).

**Source data 5.** Quantification of osteoclasts differentiated from primary macrophages.

**Source data 6.** RT-qPCR data for osteoclast differentiation from primary macrophages.

(*Figure 7A–C*). Consistent with these data, trabecular bone volume fraction measured by microCT was significantly higher in *Ksr2* KO mice than WT controls after pair-feeding (*Figure 7D*).

In the second approach, *Ksr2* was conditionally deleted in osteoblasts via *Sp7/Osterix-Cre* mice, which have been successfully used for disrupting gene function in osteoblast lineage cells (*Buettmann et al., 2019*; *Ko et al., 2021*). Since the *Osx-Cre* transgenic mice exhibit a mild skeletal phenotype (*Huang and Olsen, 2015*), we used *Osx-Cre⁺ Ksr2* floxed heterozygous mice as controls (*Figure 7E*). While neither body weight nor percent fat was different between *Osx-Cre⁺ Ksr2* floxed heterozygous (control) and homozygous (conditional KO [cKO]) mice, femoral BMD was significantly increased in the *Ksr2* cKO mice compared to control mice (*Figure 7F–H*). In sync, microCT analyses of distal femoral metaphyseal secondary spongiosa of cKO mice exhibited similar osteal gains relative to controls, as observed between global KO and WT mice (*Figure 7I–O*). Therefore, this data indicates that gains in bone mass can be regulated autonomously by KSR2 expressed in bone, independent of the centrally regulated effects of *Ksr2* in the hypothalamus.

### *Ksr2* affects osteoblast differentiation through mTOR signaling

Mechanistically, *Ksr2* has been shown to regulate changes in visceral fat by multiple mechanisms in the hypothalamus including AMPK and mTOR signaling (*Figure 8I*; *Costanzo-Garvey et al., 2009*; *Pearce et al., 2013*; *Revelli et al., 2011*). In the ST2 mouse stromal cell line, KSR2 OE inhibited *Alpl* expression in both normal and high glucose media as well as in the presence or absence of insulin treatment (*Figure 8A*).

In ST2 cells transduced with GFP control vector, both insulin and IGF-1 promoted a marked increase in the activated form of phosphorylated (pRPS6) relative to total RPS6 compared with vehicle-treated control cells, as expected. By contrast, pRPS6 activation was significantly reduced in KSR2 OE ST2 cells (*Figure 8B and C*). AMPK phosphorylation was unaffected by *Ksr2* OE in ST2 cells (data not shown). mTOR is known to promote osteoblast differentiation in ST2 cells (*Chen et al., 2014b*), and treatment of these cells under osteoblast differentiation conditions with the classic mTOR inhibitor, rapamycin, significantly blocked differentiation measured by ALPL staining (*Figure 8D*). To further determine whether *Ksr2* signals through mTOR in stromal cells during osteoblast differentiation, ST2 cells were knocked down with *Ksr2* shRNA or a nonspecific control shRNA, and osteoblast differentiation was tested with either rapamycin or vehicle control and evaluated after 48 hr for expression of *Ibsp*, *Sp7*, or *Vegfa* by RT-qPCR. Increased expression of bone formation markers by lentiviral *Ksr2* shRNA-treated cultures is abolished by rapamycin treatment (*Figure 8E*). These results indicate that *Ksr2* regulates osteoblast differentiation via mTOR activation.

To determine the downstream targets of KSR2/mTOR, we measured expression levels of Notch and hypoxia signaling genes (*Bjedov and Rallis, 2020*; *Frey et al., 2014*; *Huang et al., 2015*) in ST2 cells overexpressing KSR2 or GFP. We found that Ksr2 OE reduced the expression of hypoxia signaling targets (*Vegfa*, *Slc2a1*, *Pgk1*) but did not affect Notch targets (*Hey1*, *Hey2*), (*Figure 8F and G*). Accordingly, crucial hypoxia markers (*Hif1a*, *Vegfa*) were increased in the bones of 12-week-old *Ksr2* KO mice, but Notch targets were not changed (*Figure 8H*).

## Discussion

The impact of obesity on bone health is an area of significant concern. While obesity is known to exert complex effects on bone mass and skeletal fragility, the mechanisms by which obesity influences bone metabolism are not well understood. In this study, we used a *Ksr2* KO genetic mouse model to investigate the relationship between obesity and bone health. We found that the distal

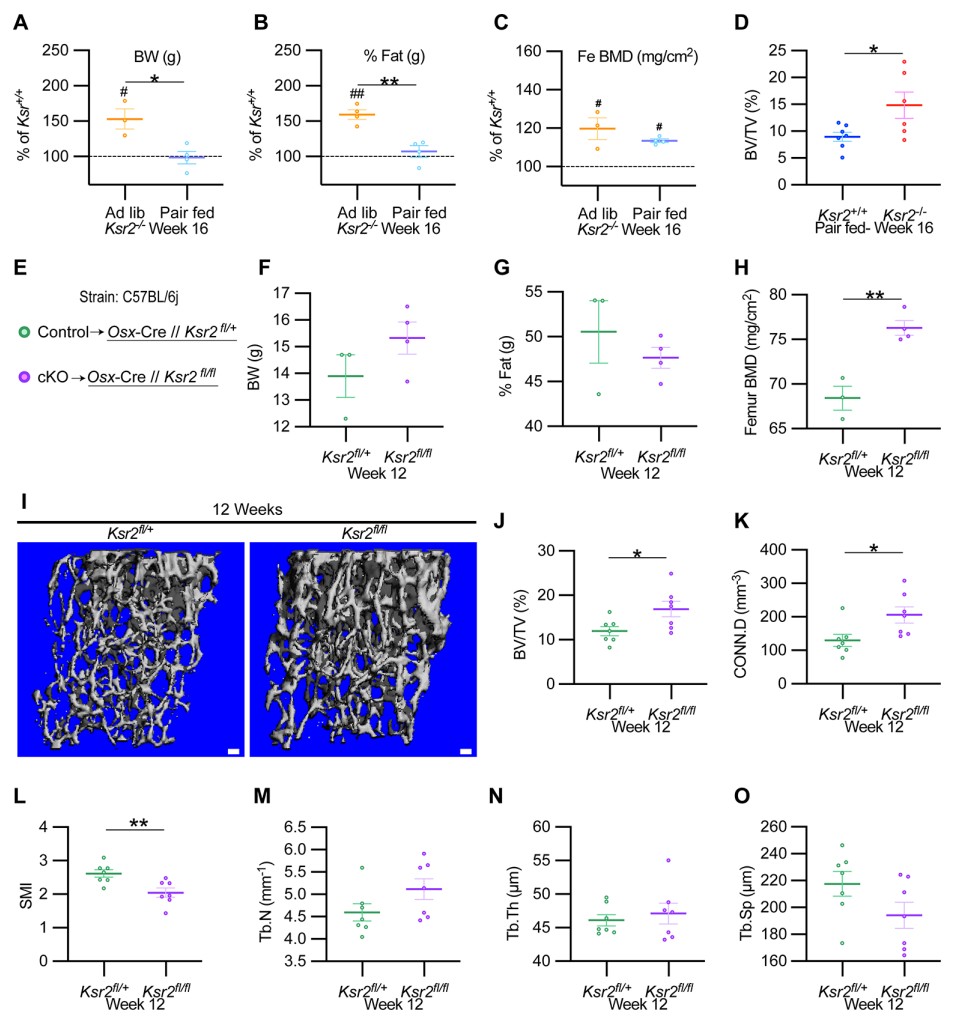

**Figure 7.** *Ksr2* regulates femoral trabecular bone autonomously. (**A–C**) Pair-feeding experiments reveal that gains in mineral density are acquired independently of eating-induced weight gains in knockout (KO) mice fed at will (Ad lib) or pair-fed according to the amount eaten by wild-type (WT) mice. Panels (**A–C**) are represented as a percentage relative to WT. BW, body weight; Fe BMD, femur bone mineral density; BV/TV, bone volume/total volume (n = 3–6/group). (**D**) BV/TV from femoral metaphysis of WT and KO at end of pair-feeding. (**E**) Conditional knockout strategy. (**F–H**) Differences between control (*Ksr2^{fl/+}*) and conditional knockout, cKO (*Ksr2^{fl/fl}*), mice in percent body fat (**F**), body weight (**G**), and femur bone mineral density (**H**) (n = 3–4/group). (Note: **B, C, G, H** reflect dual-energy X-ray absorptiometry measurements.) (**I**) Representative 3D micro-computed tomography (microCT) reconstruction images of distal femoral metaphysis in control and cKO mice at 12 weeks of age, revealing increased trabecular bone in cKO mice. Scale bar: 100 μm. (**J–O**) MicroCT measurements from the trabecular bone as represented in panel (**I**) (n = 7 mice per group; mixed genders). CONN.D, connectivity density; SMI, structural model index; Tb.N, trabecular number; Tb.Th, trabecular thickness; Tb.Sp, trabecular spacing. Statistics were analyzed by two-tailed Student's *t*-test, and graphed lines represent the mean ± SEM, *$p<0.05$, **$p<0.005$ for comparisons between groups labeled on the x-axis. In panels (**A–C**), significance between *Ksr2* KO and WT for a given condition is represented by #$p<0.05$ or ##$p<0.005$.

The online version of this article includes the following source data for figure 7:

**Source data 1.** X-ray measurements of *Ksr* knockout mice after pair-feeding experiments and micro-computed tomography (microCT) of femur metaphysis in pair-fed mice.

**Source data 2.** X-ray measurements of osteoblast-specific *Ksr2*-conditional knockout mice.

**Source data 3.** Micro-computed tomography (microCT) measurements of distal femoral metaphysis from osteoblast-specific *Ksr2*-conditional knockout mice.

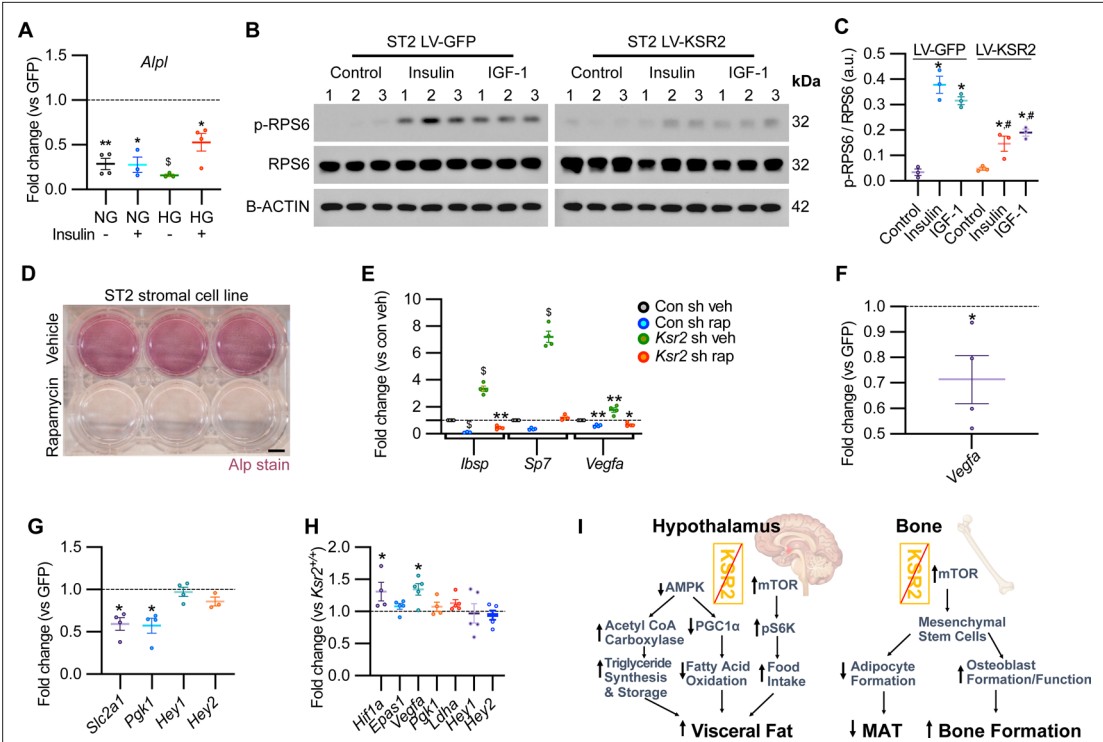

**Figure 8.** KSR2 promotes osteoblast differentiation through mTOR signaling affecting *Hif1a* and *Vegfa,* but not *Notch* signaling in the process. (**A**) RT-qPCR for *Alp* from ST2 stromal cells overexpressing KSR2 or GFP by lentivirus (LV) on day 3 in normal glucose (NG) or high glucose (HG) without (-) or with (+) insulin. (**B**) Western blot for ST2 cells overexpressing GFP or KSR2 after 30 min in vehicle control, 100 μg/ml Insulin, or 100 ng/ml IGF-1, for the mTOR response target phosphorylated p-RPS6, total RPS6, or loading control, β-actin; 1,2,3 indicate biological replicates. (**C**) Quantification of WB comparing p-RPS6/RPS6 ratios. Within-group (*), between-group (#) comparisons (n = 3/group). (**D**) Representative image of ALP activity for ST2 stromal cells in osteoblast differentiation conditions on day 7 treated with vehicle (top row) or 10 nM rapamycin (bottom row). Scale bar: 10 mm. (**E**) RT-qPCR quantification of ST2 cells transduced with empty vector control (Con) or *Ksr2* shRNA and treated with either vehicle or 10 nM rapamycin following 48 hr of osteoblast differentiation (n = 4/group). (**F, G**) RT-qPCR from ST2 stromal cells with KSR2 overexpression following 72 hr of osteoblast differentiation, plotted as a function of level detected in GFP controls (n = 4/group). (**H**) RT-qPCR on genes related with hypoxia or Notch signaling on RNA extracted from whole femurs of 12-week-old wild-type (WT) or *Ksr2* knockout (KO) mice. Values represent fold change for KO relative to WT (set to 1, dashed line). (**I**) Model diagram summarizing results where high levels of KSR2 lead to low levels of mTOR activity, resulting in low bone density, while the absence of KSR2 results in high levels of mTOR activity, resulting in high bone density. All statistics analyzed by two-tailed Student's *t*-test, graphed lines represent mean ± SEM. #$p < 0.05$, *$p < 0.05$, **$p < 0.005$, \$$p < 10^{-6}$.

The online version of this article includes the following source data for figure 8:

**Source data 1.** RT-qPCR data for ST2 stromal cells following osteoblast differentiation in low glucose or high glucose and either no insulin or with insulin.

**Source data 2.** Images of Western blot film used in *Figure 8B*.

**Source data 3.** Original Western blot scan of RPS6.

**Source data 4.** Original Western blot scan of β-actin.

**Source data 5.** Quantification of Western blot data.

**Source data 6.** RT-qPCR data for ST2 stromal cells with *Ksr2* shRNA vs. control shRNA, following osteoblast differentiation in the absence or presence of rapamycin.

**Source data 7.** RT-qPCR data for ST2 stromal cells with *Ksr2* OE vs. GFP, following osteoblast differentiation in the absence or presence of rapamycin.

**Source data 8.** RT-qPCR data of 12-week-old *Ksr2* knockout versus wild-type femur (hypoxia and Notch pathway-related genes).

femoral metaphyseal trabecular bone was considerably denser in *Ksr2* KO compared to littermate WT controls in two genetic backgrounds, while deletion of *Ksr1* did not affect trabecular bone mass. At the cellular level, we found by histomorphometric analysis, ex vivo, and in vitro studies via *Ksr2* gain and loss of function that KSR2 is a negative regulator of trabecular bone formation that may act by controlling differentiation of mesenchymal stem cells to the osteoblast or adipocyte fate. By contrast,

KSR2 did not significantly affect osteoclast formation or functions. Moreover, our data from *Ksr2* KO mice that were pair-fed similar to WT mice and osteoblast-specific *Ksr2* cKO mice revealed that *Ksr2* controls osteoblast differentiation autonomously. At the molecular level, our findings demonstrate that *Ksr2* negatively regulates osteoblast differentiation by repressing mTOR activity (*Figure 8I*). Since global *Ksr2* KO mice exhibited delayed fracture healing with an arrest at endochondral ossification, as observed in obese humans with T2D, and since KSR2 genetic polymorphisms are linked to obesity, an understanding of the cellular and molecular pathways of KSR2 regulation of mesenchymal stem cell fate may have important ramifications in promoting bone health in obese individuals.

Genetic differences between mouse strains are known to affect their susceptibility to gain excess weight, with the C57BL/6J strain being more vulnerable, and therefore used more often than DBA/1LacJ to study diet-induced weight gain (*Garofalo et al., 2003*; *Linder, 2006*). Also, the impact of genetic background on gene deletion phenotypes is well established (*Doetschman, 2009*; *Linder, 2006*). In accord, we noticed that *Ksr2* KO mice in the DBA background lived longer than those in the C57 background (data not shown). Since *Ksr2* deletion results in weight gain and increased femoral bone mass in both strains, it is highly likely that *Ksr2* also regulates bone formation in other genetic backgrounds. This warrants further investigation in other mammals and, in particular, humans. Furthermore, our data show that disruption of the *Ksr2* gene had no significant effect on trabecular bone mass in the vertebrae. These data are consistent with what is known in the literature that the heritability of BMD varies across skeletal sites (*Kemp et al., 2014*; *Rowe et al., 2018*) and differences in mechanisms that regulate bone accretion in long bones versus vertebrae.

Other monogenetic models that lead to obesity/T2D defects have also investigated bone phenotypes. In particular, knockouts of different genes in the leptin-melanocortin feedback loop that signals satiety in the hypothalamus generally result in obesity, but although the neural circuits are unidirectional, both anabolic and catabolic effects have been observed in bone. For instance, while leptin KO mice result in reduced femur length and BMD (*Steppan et al., 2000*; *Wang et al., 2007a*), they have increased vertebral bone mass (*Ducy et al., 2000*), consistent with the idea that genes can have distinct effects in different anatomical regions. By contrast, *Mc4r* and *Npy1r* knockouts result in increased femoral BMD (*Ahn et al., 2006*; *Baldock et al., 2007*; *Braun et al., 2012*), while *Mc3r* knockouts have reduced femur length and BMD (*Lee et al., 2016*). Many of these genes, such as *Npy1R* and *Mc4r,* are expressed in both hypothalamic neurons and osteoblasts (*Baldock et al., 2007*; *Zhong et al., 2005*), which may be partly responsible for the complex skeletal phenotypes seen in these mice. Similarly, *Ksr2* is expressed in the hypothalamus, and in this study, we found that it is also expressed and functional in cells of the osteoblastic lineage. Although our pair-feeding and conditional knockout studies define a role for KSR2 function in bone that can be dissociated from its hypothalamic function, *Ksr2* global knockout and likely humans with KSR2 genetic polymorphisms have malfunctions of both hypothalamic KSR2 regulated food intake as well as bone KSR2 regulated osteoblast formation. In future studies, we will address whether *Ksr2* plays similar or different roles in other skeletal sites and whether conditional hypothalamic deletion of *Ksr2* has any effect on bone physiology.

An alternative means to study the effect of obesity on bone is provided by diet-induced obesity models. Generally, these mice result in excess body fat, with reduced trabecular bone mass at the expense of increased MAT (*Bonnet et al., 2014*; *Scheller et al., 2016*; *Tencerova et al., 2018*). This would suggest excess body adipocytes, which secrete adipokines that are known to influence different aspects of bone maintenance by regulating the differentiation or function of BMSCs, osteoblasts, or osteoclasts. Consistent with the idea that adipocyte-derived factors regulate osteoblast and osteoclast functions are the findings that bone mass is increased under conditions of generalized reduction in adipose tissue, as in the case of congenital lipodystrophy (*Zou et al., 2019*). However, the local secretion of adiponectin, an adipokine, in MAT appears to provide a stronger influence in this model (*Zhong et al., 2020*). Consistently, we did detect a decrease in adiponectin expression in the femurs of *Ksr2* KO mice (data not shown). Nevertheless, this indicates that adipocytes can influence bone homeostasis by both systemic and local signals.

Leptin is another well-known adipokine secreted by adipocytes that is increased in obese animals and is known to regulate bone formation. However, a significant role for adipocyte-derived leptin in mediating the gains in trabecular bone mass in *Ksr2* KO mice does not seem likely. We previously found that leptin is increased in serum of *Ksr2* KO mice, but leptin resistance was not causative of

weight gains in these mice (*Costanzo-Garvey et al., 2009*; *Revelli et al., 2011*). Here we show that *leptin* mRNA is increased in body fat depots but believe this is also not responsible for increased bone mass. Our data shows that pair-fed *Ksr2* KO mice and osteoblast-specific conditional *Ksr2* KO mice do not gain excess adipocytes, yet still resulted in gains in trabecular bone. Since leptin is produced by fat, which did not change in either of these conditions, these results suggest that gains in systemic leptin produced by excess fat may not be a causative factor in bone mass accretion when *Ksr2* is deleted. Moreover, while we did see a reduction in *leptin* mRNA expression in femurs, in isolation this might be expected to reduce bone mass, given that in vitro and in vivo reports indicate that leptin is anabolic to limb bones (*Astudillo et al., 2008*; *Gordeladze et al., 2002*; *Steppan et al., 2000*; *Wang et al., 2007a*). While it has been reported that *leptin* mRNA is higher in visceral adipose tissue than BMAT (*Liu et al., 2011*), in the absence of KSR2, it is reduced even further, yet this does not result in bone loss.

Interestingly, MAT is significantly reduced in the long bones of *Ksr2* KO mice compared to WT mice as revealed by osmium tetroxide microCT evaluation, as well as histological analyses of adipocyte numbers. By contrast, MAT has been shown to be increased in mice with disruption of leptin, leptin receptor, as well as mice fed with high-fat diets, all three models that show reduced femoral trabecular bone mass (*Hamrick et al., 2004*; *Tencerova et al., 2018*; *Yue et al., 2016*). These findings, together with the known fact that mesenchymal stem cells represent common precursors of both osteoblasts and adipocytes, raise the possibility that KSR2 might modulate the switch between osteoblast and adipocyte differentiation produced by mesenchymal stem cells, and, thereby, bone formation and MAT. Further studies are required to determine the cause-and-effect relationship between changes in *Ksr2* expression and regulation of mesenchymal stem cell differentiation. In this regard, a recent study demonstrated that complement factor D/adipsin from bone marrow adipocytes regulates bone marrow stromal cell fate determination through activation of the complement system (*Aaron et al., 2021*). We, therefore, examined whether the expression of adipsin was altered in the bones of *Ksr2* KO mice and found reduced expression of adipsin in both body fat and femoral adipocytes. The issue of whether KSR2 regulates adipsin expression directly or indirectly via other factors remains to be established. While Aaron et al. demonstrated that adipsin is a downstream target of PPARG, *Pparg* transcription was only mildly reduced in *Ksr2* KO bones, thus raising the possibility that KSR2 might regulate adipsin expression independently of PPARG. BMSC fate decision is regulated by multiple factors, including many known, and potentially unknown, growth factors and hormones (*Chen et al., 2016*). Interestingly, non-endocrinological neural regulation mechanisms also contribute to MAT formation (*Zhang et al., 2021*). Also, factors secreted by osteoblasts such as osteopontin have been shown to locally regulate bone MAT (*Chen et al., 2014a*). Therefore, KSR2 might modulate BMSC fate by regulation of adipsin expression, as well as the response to signaling by other factors, which await further investigation.

Other factors that might affect bone physiology in obese and T2D conditions are inflammatory cytokines. While adipocytes are known to contribute to increased levels of pro-inflammatory cytokines such as TNF, Il6, Il17, and Tnfsf11/RANKL (*Benova and Tencerova, 2020*; *Kawai et al., 2021*) during certain pathological states that can promote increased osteoclastogenesis and resorptive activity, we did not see changes in bone resorption in *Ksr2* KO mice. However, increased levels of these inflammatory cytokines could be responsible for the altered fracture healing in *Ksr2* KO mice. Diet-induced obesity models have reported reduced callus bone volume and increased marrow adiposity, possibly due to a faster rate of callus resorption (*Brown et al., 2014*), producing bones with microstructural deficits in collagen matrix and increased advanced glycation end products (*Khajuria et al., 2020*). Although some studies find smaller callus in obesity/T2D fracture callus (*Brown et al., 2014*), others have also observed increased callus size in the fracture callus of DIO-obesity/T2D model fractures, with increased hypertrophic chondrocytes, and delayed fracture healing, similar to the results reported here (*Marin et al., 2021*). It is possible that fractures using an intramedullary pin are not as stabilized in the *Ksr2* KO mice as that of WTs because of increased body weight, thus leading to a larger less dense callus, a phenomenon frequently seen in non-stabilized human fractures. Further time-course studies are needed to determine the cause for the delayed remodeling of fracture callus in the *Ksr2* KO mice, and whether the healed bones in *Ksr2* KO mice are mechanically weaker than the healed bones of control mice. Regardless, the diabetic state results in a deranged inflammatory condition that is believed to affect the vascular system by the production of advanced glycation end products

and multiple factors that may delay fracture healing (*Marin et al., 2018*), and we show here that loss of *Ksr2* may not be sufficient to impart improved fracture healing for diabetics. In the future, we will determine whether increased general adiposity is the cause of delayed fracture healing in global *Ksr2* KO mice by evaluating whether fracture healing is affected in mice with conditional disruption of *Ksr2* in osteoblasts. Nevertheless, bones of *Ksr2* KO mice were less resistant to fracture, in agreement with the observation in obese/T2D humans that present with increased bone mass and fracture susceptibility (*Greco et al., 2015*; *Ma et al., 2018*; *Moseley, 2012*; *Oei et al., 2013*).

mTOR regulation of osteoblast formation remains controversial as both positive and negative associations have been reported (*Chen et al., 2014b*; *Martin et al., 2010*; *Martin et al., 2015*; *Xian et al., 2012*; *Yeo et al., 2021*). Our in vitro results show that forced expression of KSR2 reduces mTOR signaling, while knockdown of *Ksr2* promotes induction of osteogenic factors, but not when mTOR signaling is inhibited by rapamycin. Moreover, both in vitro and in vivo results suggest that KSR2 and mTOR affect hypoxia but not Notch signaling genes. The anabolic effects of hypoxia signaling on bone mass and vasculature are well established (*Mohan and Kesavan, 2022*; *Shen et al., 2009*; *Wan et al., 2010*; *Wang et al., 2007b*; *Wolf et al., 2022*). Thus, whether the increased hypoxia signaling pathway observed in bones of *Ksr2* KO mice contributes to the increased trabecular bone mass remains to be established. Thus, this work implicates mTOR as a positive effector of osteoblast differentiation that can be regulated by KSR2 (*Figure 8I*). Future studies are needed to determine how KSR2 regulates mTOR signaling biochemically, whether KSR2 regulates BMSC fate decision via mTOR in either the mTORC1 or mTORC2 complex, which is reportedly one means of affecting BMSC fate regulation (*Martin et al., 2015*; *Sen et al., 2014*), and whether osteoblast-specific deletion of mTOR in *Ksr2* KO mice will reverse the bone gains in *Ksr2* KO mice. Studies have shown that KSR2 is a scaffold protein that interacts with RAF and MEK to facilitate activation of ERK/MAPK module (*Roy et al., 2002*). In addition, a recent study demonstrated that KSR1 and KSR2 when expressed at high levels can activate the MAPK pathway-independent of RAS (*Paniagua et al., 2022*). Based on the published data that ERK effects on osteoblasts are mediated via mTOR signaling (*Kim et al., 2022*), it is possible that KSR2 effects on bone are via MAPK-mediated regulation of mTOR signaling. Our future studies will investigate this possibility.

In summary, our investigation of bones in *Ksr2* knockout genetic mouse models resulted in the identification of a novel animal model in which the obesity/T2D condition coincides with increased appendicular bone mass. Since KSR2 genetic polymorphisms are linked to obesity/T2D in humans, our full understanding of how KSR2 differentially regulates general tissue adiposity versus bone marrow adiposity could lead to the identification of novel therapeutic strategies to promote bone health in humans with obesity/T2D.

## Methods

**Key resources table**

| Reagent type (species) or resource | Designation | Source or reference | Identifiers | Additional information |
|---|---|---|---|---|
| Antibody | Anti-COL10A1 (rabbit polyclonal) | Abcam | Cat# ab58632; RRID:AB_879742 | IF (1:100) |
| Antibody | Anti-IBSP (rabbit polyclonal) | Dr. Reny Franceschi, University of Michigan | | IF (1:100) rabbit serum |
| Antibody | Anti-SPP1 (rabbit polyclonal) | Kerafast | Cat# ENH094-FP | IF (1:300) |
| Antibody | Anti-SP7/OSX (rabbit polyclonal) | Abcam | Cat# ab22552; RRID:AB_2194492 | IF (1:100) |
| Antibody | Anti-KSR2 (rabbit polyclonal) | Novus Biologicals | Cat# nbp1-83553; RRID:AB_11034779 | IF (1:100) |
| Antibody | Anti-goat IgG (H+L) (horse polyclonal) | Vector Laboratories | Cat# DI-3088; RRID:AB_2336400 | IF (1×) |
| Antibody | Anti-rabbit IgG (horse polyclonal) | Vector Laboratories | Cat# DI-1794; RRID:AB_2336784 | IF (1×) |

*Continued on next page*

*Continued*

| Reagent type (species) or resource | Designation | Source or reference | Identifiers | Additional information |
|---|---|---|---|---|
| Other | DAPI stain | Invitrogen | Cat# D1306; RRID:AB_2629482 | (1 µg/ml) |
| Antibody | Anti-S6 ribosomal protein (rabbit monoclonal) | Cell Signaling Technology | Cat# 2217; RRID:AB_331355 | WB (1:1000) |
| Antibody | Anti-phospho S6 ribosomal protein (rabbit polyclonal) | Cell Signaling Technology | Cat# 2215; RRID:AB331682 | WB (1:1000) |
| Antibody | Anti-β-actin (mouse monoclonal) | Sigma-Aldrich | Cat# A1978; RRID:AB476692 | WB (1:1000) |
| Antibody | Anti-rabbit IgG HRP (goat polyclonal) | Sigma-Aldrich | Cat# A9169; RRID:AB_258434 | WB (1:15,000) |
| Antibody | Anti-mouse IgG HRP (rabbit polyclonal) | Novus Biologicals | Cat# NB720-H; RRID:AB_524513 | WB (1:15,000) |
| Other | PINP-ELISA kit | Immunodiagnostic Systems | Cat# AC-33F1 | |
| Other | CTX-1-ELISA kit | Immunodiagnostic Systems | Cat# AC-06F1 | |
| Cell line (*Mus musculus*) | ST2 | ATCC | Cat# PTA-10431 | |
| Recombinant DNA reagent | pRRLin-CPPT-SFFV-E2A-GFP-wpre | Addgene | 12252; RRID:Addgene_12252 | |
| Recombinant DNA reagent | pCDNA3.1 KSR2 | Addgene | 25968; RRID:Addgene_25968 | |
| Other | shRNA for Ksr2 (*Mus musculus*) | MilliporeSigma | TRCN0000378606 | Refseq target: NM_001114545 |
| Other | shRNA control (*Mus musculus*) | MilliporeSigma | SHC002V | |

## Mice

Femoral bones of mice in the C57BL/6J-Tyr[c-Brd] × 129[SvEvBrd] hybrid background were transferred from Lexicon Pharmaceuticals to the Veteran's Affairs Loma Linda Healthcare System (VALLHS) and analyzed at VALLHS. *Ksr2*[+/-] mice in the DBA/1LacJ were transferred from the University of Nebraska to the VALLHS and maintained by inbreeding for experimentation and further analysis. *Ksr2*-floxed mice were generated by insertion of LoxP sites flanking exon3 of *Ksr2* as described (*Guo et al., 2017*), and mated to *Sp7/Osx*-Cre mice (a kind gift from Dr. Andrew P. McMahon, University of Southern California, USA) for osteoblast-specific deletion of *Ksr2*. *Ksr1*[+/-] mice were a kind gift from Dr. Andrey S. Shaw at Washington University School of Medicine (St. Louis, USA) and were bred to purity in the C57BL/6J background for bone analysis. Mice genotyping was done by conventional tail snip PCR with DNA primers. All animals were housed at the animal facility of VALLHS (Loma Linda, CA) according to approved standards with controlled temperature (22°C) and illumination (14-hr light, 10-hr dark). Mice were fed a standard chow diet. The approved anesthetic (isoflurane) was used for anesthesia, and $CO_2$ exposure was used for euthanasia followed by cervical dislocation.

## Pair-feeding

Pair-feeding studies were performed as described (*Pearce et al., 2013*; *Revelli et al., 2011*). Mice were fed a standard chow diet throughout the experiment.

## Fractures

At 16 weeks of age, *Ksr2* KO and WT mice of mixed genders were subjected to stabilized closed femoral fracture by a modification of the three-point bending approach (*Rundle et al., 2008*). Fracture tissues were harvested at 3 weeks post-fracture for further analysis when bony callus union is expected in this model and after which fracture callus remodeling should normally complete healing.

## MicroCT

Femur lengths and trabecular and cortical bone parameters were measured on a VIVA CT40 (Scanco Medical, Bruttisellen, Switzerland) microCT system. Bones were fixed in 10% formalin overnight, washed, and imaged in 1× PBS with 55–70 kVp volts at a voxel size of 10.5 μm. Images were reconstructed using the 2D and 3D image software provided by the Scanco VIVA-CT 40 instrument (Scanco USA, Wayne, PA). For analysis of the spine, bones were sampled at the fourth lumbar (L4) vertebrae. Osmium tetroxide experiments were performed for the measurement of MAT as described (*Lindsey et al., 2019a*).

## Dual X-ray absroptiometry

Total body BMD, percent body fat, femoral BMD, and X-ray fracture images were analyzed on a Faxitron Radiography system (Hologic, Bedford, MA). Images were acquired with 20 kV X-ray energy for 10 s.

## Three-point bending strength test

Three-point bending strength test was performed as previously described (*Mohan et al., 2000*). Tibiae were fixed in 10% formalin for 3–5 days at 4°C and stored frozen in gauze moistened in PBS with 0.01% sodium azide, prior to thawing in PBS at 4°C. Samples were tested by three-point bending with the Instron DynaMight testing system (Model 8840; Instron, Canton, MA).

## Bone histomorphometry

Seven-week-old mice were injected with calcein (20 mg/kg) at 8 days and 2 days before histomorphometric measurements on week 8 as described (*Xing et al., 2013*). Calcein retaining trabeculae and tartrate-resistant acid phosphatase (TRAP)-labeled trabecular surfaces were measured in a blinded fashion with OsteoMeasure (OsteoMetrics, Decatur, GA) software.

## Histology

Mouse femurs were fixed in 10% formalin overnight, washed in PBS, decalcified in 10% EDTA (pH 7.4) at 4°C for 7 days while shaking, and embedded in paraffin for sectioning. Longitudinal sections of distal femurs were stained with alizarin red, and hematoxylin and eosin using standard procedures. Fracture calli were stained with Safranin O or acid phosphatase 5, tartrate-resistant/TRAP (Sigma-Aldrich) followed by fast green counterstain. TRAP (S387A, Sigma-Aldrich), alizarin red (A5533, Sigma-Aldrich), and alkaline phosphatase, ALP (N6125 and F3381, Sigma-Aldrich) staining of cell cultures were performed by standard procedures.

## Immunofluorescence

Longitudinal paraffin-embedded sections were processed as described (*Gomez et al., 2022*) following 1 hr antigen retrieval with 2 mg/ml hyaluronidase (Sigma-Aldrich) at 37°C. Sections were blocked in 2.5% normal horse serum and incubated overnight with primary antibodies for COL10A1 at 1:100 (ab58632, Abcam), IBSP at 1:100 (gift from Dr. Renny Franceschi, University of Michigan), SPP1 at 1:300 (ENH094-FP, Kerafast), SP7/OSX at 1:100 (ab22552, Abcam), and KSR2 at 1:100 (nbp1-83553, Novus Biologicals). Protein expression was detected by species-specific secondary antibodies (Vector Laboratories, DI-3088, and DI-1794), followed by DAPI (D1306, Invitrogen) counterstain before imaging.

## Microscopy

Epifluorescence images were obtained on a Leica Digital Microscope DMI6000B with Leica Applicate Suite X software or an Olympus FV3000 confocal microscope via FV31S-SW software. Colorimetric histological images were obtained with an Olympus DP72 camera attached to an Olympus DP72 camera through DP2-BSW software.

## ELISA

Serum levels of P1NP, and collagen type 1 C-terminal telopeptide (Ctx-1) EIA kits, all from Immunodiagnostic Systems (Gaithersburg, MD) were obtained according to the manufacturer's instructions.

## Western blot

Immunoblots were processed by standard procedures. Cells were lysed in RIPA buffer with 1 mM DTT, 1× protease inhibitor, and 1× phosphatase inhibitor cocktail (Sigma-Aldrich). Protein lysate concentrations were determined with a BCA protein assay (Thermo Scientific) and 10 µg of each lysate was boiled in 4× SDS dye, then loaded on 10% SDS-PAGE gels for immunoblotting on PVDF membranes. Membranes were blocked in 4% BSA in 1× TBS and probed with S6 ribosomal protein 1:1000 (2217, Cell Signaling Technologies), phosphor-S6 ribosomal protein 1:1000 (2215, Cell Signaling Technologies), or β-actin 1:5000 (A1978, Sigma-Aldrich). Primary antibodies were detected with goat anti-rabbit IgG-HRP (A9169, Sigma) or rabbit anti-mouse IgG-HRP (NB720-H, Novus Biologicals) at 1:15,000. Blots were detected with Immobilon Chemiluminescent HRP substrate (P90720, MilliporeSigma) and exposed on autoradiography film (1968-3057, USA Scientific). Scanned images were quantified on ImageJ software.

## Real-time quantitative PCR

RNA was extracted from adipocyte depots, bones, or cultured cells with TRI reagent (Molecular Research Center INC, TR118) according to the manufacturer's instructions and purified on silica columns with E.Z.N.A. Total RNA Kit I (R6834-02, Omega Bio-tek). Total RNA was reverse transcribed to cDNA with Oligo(dT)12–18 and Superscript IV Reverse transcriptase (18091050, Invitrogen). Real-time PCR reactions were processed on a ViiA 7 RT-PCR system (Applied Biosystems). All reactions were standardized with peptidyl prolyl isomerase A (*Ppia*) primers. Primer sequences used for RT-qPCR are listed in *Supplementary file 1*. Fold changes were calculated by the Delta Ct method.

## Cell culture

All cells were maintained in standard normoxic conditions; humidified, 37°C, 5% $CO_2$ with 1% penicillin/streptomycin (Gibco). ST2 stem/stromal cell line was obtained from the American Type Culture Collection (Manassas, VA), tested negative for mycoplasma, and were authenticated by their ability to differentiate into chondrocytic, adipocytic, and osteoblastic lineages in their respective differentiation media. For gain-of-function studies, the coding region of GFP in pRRLin-CPPT-SFFV-E2A-GFP-wpre (LV-GFP) was swapped with that of *Ksr2* from pcDNA3-*Ksr2*-flag (Addgene), producing pRRLsin-CPPT-SFFV-E2A-*KSR2*-wpre (LV-KSR2). Lentivirus (LV) plasmids were co-transfected with Pax2 and VSVG plasmids in 293T cells for LV generation as previously reported (*Lindsey et al., 2019b*). LV particles were transduced directly into ST2 or BMSCs. ST2 cells were cultured in 10% CS (Hyclone) with no AA (Life Technologies). Osteoblast differentiation was performed with 10 mM β-glycerophosphate (BGP) and 50 µg/ml AA (Sigma-Aldrich), with BGP only serving as vehicle. Glucose, insulin, rapamycin, and IGF-1 (MilliporeSigma) were added at concentrations mentioned in the text, and low glucose (LG) was 5.5 mM, while high glucose (HG) was 25 mM. Mission lentiviral transduction shRNA particles for control (SHC002V) and *Ksr2* (TRCN0000378606) were obtained from MilliporeSigma, and cells were selected in 10 µg/ml puromycin for 1 week before osteoblast differentiation.

Ex vivo culture of calvarial osteoblasts were isolated from 21-day-old C57BL/6J mice and maintained in 10% FBS (Gibco) AMEM no AA (Life Technologies), before osteoblast differentiation. BMSCs were isolated from whole femurs and tibias of 4- to 6-week-old C57BL/6J mice, while macrophages were isolated from 8-week-old C57BL/7J mice. KSR2 and GFP were overexpressed by lentivirus, without antibiotic selection. Osteoclast differentiation was performed with 30 ng/ml MCSF (R&D), and 30 ng/ml RANKL (R&D), with MCSF only serving as vehicle controls.

## Figures

Figures were assembled on Adobe Illustrator CS5. Quantitative graphs were generated on Prism v9.3.1 software (GraphPad).

## Statistics

Statistical analysis was performed by two-tailed Student's *t*-test on Excel (Microsoft Office 365) following tests for normality. Data are presented as mean ± standard error of the mean (SEM) throughout. Values were considered significant at $p < 0.05$ or less.

## Study approval

Animal studies were performed according to protocols approved by the Institutional Animal Care and Use Committee of the VALLHS (Protocol#: 0029/204).

## Acknowledgements

We would like to thank Dr. Robert Brommage for helpful critical comments on the manuscript; Dr. Andrew S Shaw and Dr. McMahon for mice, and Dr. Renny Franceschi for the BSP antibody; and Jasmine Lau, Fern Baedyananda, William Tambunan, Destiney Larkin, and Nancy Lowen for excellent technical assistance. This work was supported by grants from the National Institutes of Health (R01 AR048139 and R21 AG062866) and US Department of Veterans Affairs (101 BX005263 and Ik6 BX005381) awarded to SM. SM is a recipient of a Senior Research Career Scientist Award from the US Department of Veterans Affairs.

## Additional information

### Competing interests

David R Powell: was employed by Lexicon Pharmaceuticals, Inc, at the time some of these studies were performed and may own common stock or have been granted stock options. Subburaman Mohan: Reviewing editor, *eLife*. The other authors declare that no competing interests exist.

### Funding

| Funder | Grant reference number | Author |
| --- | --- | --- |
| National Institute of Arthritis and Musculoskeletal and Skin Diseases | R01 AR048139 | Subburaman Mohan |
| National Institute on Aging | R21 AG062866 | Subburaman Mohan |
| U.S. Department of Veterans Affairs | 101 BX005263 | Subburaman Mohan |
| U.S. Department of Veterans Affairs | Ik6 BX005381 | Subburaman Mohan |
| U.S. Department of Veterans Affairs | Senior Research Career Scientist Award | Subburaman Mohan |

The funders had no role in study design, data collection and interpretation, or the decision to submit the work for publication.

### Author contributions

Gustavo A Gomez, Data curation, Formal analysis, Investigation, Methodology, Writing - original draft, Writing – review and editing; Charles H Rundle, Data curation, Investigation, Methodology, Writing – review and editing; Weirong Xing, Formal analysis, Investigation, Methodology, Writing – review and editing; Chandrasekhar Kesavan, Investigation, Methodology, Writing – review and editing; Sheila Pourteymoor, Data curation, Formal analysis, Methodology; Robert E Lewis, Resources, Writing – review and editing; David R Powell, Resources, Investigation, Writing – review and editing; Subburaman Mohan, Conceptualization, Resources, Data curation, Formal analysis, Supervision, Funding acquisition, Investigation, Methodology, Writing - original draft, Project administration, Writing – review and editing

### Author ORCIDs

Gustavo A Gomez  http://orcid.org/0000-0001-9294-4276
Robert E Lewis  http://orcid.org/0000-0002-3616-2971
David R Powell  http://orcid.org/0000-0003-0656-0124
Subburaman Mohan  http://orcid.org/0000-0003-0063-986X

### Ethics

All animals were housed at the animal facility of VALLHS (Loma Linda, CA, USA) according to approved standards with controlled temperature (22°C) and illumination (14-hour light, 10-hour dark). Mice were fed a standard chow diet. The approved anesthetic (isoflurane) was used for anesthesia, and $CO_2$ exposure was used for euthanasia followed by cervical dislocation. Animal studies were performed

according to protocols approved by the Institutional Animal Care and Use Committee of the VALLHS ( protocol#: 0029/204).

## Decision letter and Author response

Decision letter https://doi.org/10.7554/eLife.82810.sa1
Author response https://doi.org/10.7554/eLife.82810.sa2

---

## Additional files

### Supplementary files
• Supplementary file 1. Real-time quantitative PCR primers used in this study. Methods Table 1.
• MDAR checklist

### Data availability
All data generated or analysed during this study are included in the manuscript and supporting file; Source Data files have been provided for all figures.

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
