## [Editor Report]

This study represents an important advance in connecting bone biology and metabolic functions. It implicates *Ksr2* as a key regulator of the switch between adipocytes and osteoblasts that arise from a common precursor. Besides being an actionable target for obesity and osteoporosis, the study reaffirms and provides mechanistic data relating to the human genetic findings on KSR2 variants in metabolic regulation.

---

## [Decision Letter]

**Decision letter after peer review:**

Thank you for submitting your article "Contrasting effects of *Ksr2*, an obesity gene, on trabecular bone volume and bone marrow adiposity" for consideration by *eLife*. Your article has been reviewed by 2 peer reviewers, one of whom is a member of our Board of Reviewing Editors, and the evaluation has been overseen by Mone Zaidi as the Senior Editor. The reviewers have opted to remain anonymous.

Essential revisions:

1) The paper clearly shows evidence for KSR2 regulation of mTOR signaling. However, one question that is not addressed relates to how a scaffolding protein such as KSR2 regulates mTOR signaling. Please address the potential connection between KSR2 and mTOR.

2) Serum assays of bone turnover markers were done using serum from 6-month-old mice that reveal no differences. Are there measurements of serum bone formation markers in younger mice?

3) The paper proposes an interesting mechanism involving adipsin for KSR2 regulation of marrow adipose tissue. Are there other candidates that could be involved?

4) If there are data, the inclusion of circulating serum markers like the adiponectin data would be interesting. Similarly, any ex vivo experiments from the conditional knock out which presumably does not have the same serum marker changes would be interesting to include and discuss, if available.

*Reviewer #1 (Recommendations for the authors):*

1) The authors clearly show evidence for KSR2 regulation of mTOR signaling. However, one question that is not addressed relates to how a scaffolding protein such as KSR2 regulates mTOR signaling. The authors should address the potential connection between KSR2 and mTOR.

2) Serum assays of bone turnover markers were done using serum from 6-month-old mice that reveal no differences. Have the authors measured serum bone formation markers in younger mice?

3) The authors propose an interesting mechanism involving adipsin for KSR2 regulation of marrow adipose tissue. Are there other candidates that could be involved?

---

## [Author Response]

Essential revisions:1) The paper clearly shows evidence for KSR2 regulation of mTOR signaling. However, one question that is not addressed relates to how a scaffolding protein such as KSR2 regulates mTOR signaling. Please address the potential connection between KSR2 and mTOR.

Studies have shown that KSR2 is a scaffold protein that interact with RAF and MEK to facilitate activation of the ERK/MAPK module (PMID:11850406). In addition, a recent study demonstrated that KSR1 and KSR2 when expressed at high levels can activate the MAPK pathway-independently of RAS (PMID:35313064). Based on the published data that ERK effects on osteoblasts is mediated via mTOR signaling (PMID:35975983), it is possible that KSR2 effects on bone are via MAPK mediated regulation of mTOR signaling. Our future studies will investigate this possibility.

2) Serum assays of bone turnover markers were done using serum from 6-month-old mice that reveal no differences. Are there measurements of serum bone formation markers in younger mice?

We agree, given that serum of younger mice is more relevant to the data presented, we obtained measurements from mice at 8 weeks and reported the relevant data in place of that from 6-month-old mice in the manuscript. Our new data show that serum levels of P1NP (bone formation marker) was increased while that of CTX (bone resorption marker) was unchanged in the Ksr2 knock out mice. The corresponding text was modified.

3) The paper proposes an interesting mechanism involving adipsin for KSR2 regulation of marrow adipose tissue. Are there other candidates that could be involved?

Definitely. We focused our attention on adipsin because the recent publication clearly shows it has a convincing effect on regulating bone marrow stem cell fate towards adipocytes at the expense of osteoblasts. The bone marrow stem cell fate decision is regulated by multiple factors including many known, and potentially unknown, growth factors and hormones (PMID:26868907). Interestingly, nonendocrinological neural regulation mechanisms also contribute to marrow adipose tissue formation (PMID:33766429). Also, factors secreted by osteoblasts such as Osteopontin have been shown to locally regulate bone marrow adipose tissue (PMID:24123709). Therefore, KSR2 might modulate bone marrow stem cell fate by regulation of adipsin expression, as well as the response to signaling by other factors, which await further investigation.

4) If there are data, the inclusion of circulating serum markers like the adiponectin data would be interesting. Similarly, any ex vivo experiments from the conditional knock out which presumably does not have the same serum marker changes would be interesting to include and discuss, if available.

Thank you for this suggestion. Unfortunately, we do not have any serum left to perform additional measurements on adipocyte markers and it will take considerable amount of time to generate additional mice needed for this work. We are in the process of generating additional mice to address further questions on how KSR2 regulates the fate of bone marrow stem cells towards adipocytes/osteoblasts and we hope to include serum measurements of adipocyte markers during the course of these investigations.